# Genomic diversity landscapes in outcrossing and selfing *Caenorhabditis* nematodes

**Anastasia A. Teterina**[1,2]*, **John H. Willis**[1], **Matt Lukac**[1], **Richard Jovelin**[3], **Asher D. Cutter**[3], **Patrick C. Phillips**[1]

**1** Institute of Ecology and Evolution, University of Oregon, Eugene, Oregon, United States of America, **2** Center of Parasitology, Severtsov Institute of Ecology and Evolution RAS, Moscow, Russia, **3** Department of Ecology and Evolutionary Biology, University of Toronto, Toronto, Ontario, Canada

* teterina.anastasia@gmail.com

**Data Availability Statement:** All genomic data generated in this study is available in the Sequence Read Archive (SRA) at NCBI under the accession numbers PRJNA800818 and PRJNA799839. Other

## Abstract

*Caenorhabditis* nematodes form an excellent model for studying how the mode of reproduction affects genetic diversity, as some species reproduce via outcrossing whereas others can self-fertilize. Currently, chromosome-level patterns of diversity and recombination are only available for self-reproducing *Caenorhabditis*, making the generality of genomic patterns across the genus unclear given the profound potential influence of reproductive mode. Here we present a whole-genome diversity landscape, coupled with a new genetic map, for the outcrossing nematode *C. remanei*. We demonstrate that the genomic distribution of recombination in *C. remanei*, like the model nematode *C. elegans*, shows high recombination rates on chromosome arms and low rates toward the central regions. Patterns of genetic variation across the genome are also similar between these species, but differ dramatically in scale, being tenfold greater for *C. remanei*. Historical reconstructions of variation in effective population size over the past million generations echo this difference in polymorphism. Evolutionary simulations demonstrate how selection, recombination, mutation, and selfing shape variation along the genome, and that multiple drivers can produce patterns similar to those observed in natural populations. The results illustrate how genome organization and selection play a crucial role in shaping the genomic pattern of diversity whereas demographic processes scale the level of diversity across the genome as a whole.

## Author summary

The mode of reproductive exchange among individuals has a profound effect on genetic diversity. In self-reproducing organisms, absence of genetic interchange between individuals reduces the effective population size and increases linkage among segregating sites at different genes, leading to lower diversity than outcrossing species. *Caenorhabditis* nematodes offer an exceptional system for studying the genomic effects of different systems of mating. While selfing species such as *C. elegans* have been studied, we present the first recombination map and genome-wide landscape of polymorphism for an outcrossing member of the genus, *C. remanei*. We find that, similar to *C. elegans*, *C. remanei* has high recombination rates on chromosome arms and low rates in central regions. The genomic

data reanalyzed in this study is available under the accession number PRJNA549503. The code used in this work is available at https://github.com/phillips-lab/CR_CE_popgen. Supporting datasets for manuscript available at https://doi.org/10.6084/m9.figshare.23826486.

**Funding:** This study was funded by a Natural Sciences and Engineering Research Council of Canada (NSERC) Discovery Grant to ADC, a National Institute of General Medical Sciences of the National Institutes of Health grant (R01 GM096008) to ADC and PCP, and a National Institutes of Health (NIH) MIRA award (R35 GM131838) to PCP. The funders had no role in study design, data collection and analysis, decision to publish, or preparation of the manuscript.

**Competing interests:** The authors have declared that no competing interests exist.

diversity landscapes of these species are qualitatively similar, with higher diversity in the regions of higher recombination. However, *C. remanei* exhibits tenfold greater diversity than *C. elegans* due to their much larger effective population size and the decreased impact of linked selection as an outcrossing species. We use evolutionary simulations to show the influence of genomic and demographic processes work on these patterns. This work illustrates how understanding complex interactions among genetics, genomics, and reproduction is fundamental to describing patterns of genetic variation within natural populations.

## Introduction

Population genomics aims to infer the evolutionary forces and historical processes that have shaped genetic variation within species while considering the effects of a range of factors such as natural selection, patterns of reproduction, genome functional organization, mutational and recombinational landscapes, as well as spatial and temporal population dynamics and demographic history. As most of these factors act in combination, it can be challenging to infer evolutionary history using DNA sequence information within natural populations. While many population genetic models incorporate several of these factors [1–4], the enormous complexity of the problem means that there is no single analytical model that encompasses all of these interconnected processes. Even as the scale and quality of individual-level genomic data within natural populations continue to mount, a great deal of work remains to fully integrate the ways that mechanistic genetic processes manifest patterns at the whole-genome level in combination with evolutionary processes operating within and between populations and species.

Given this challenge, one promising and integrative approach is to use current knowledge about the genetics of a given species alongside evolutionary simulations of a variety of evolutionary scenarios [5–10] so as to generate a series of null models for hypothesis testing of empirical data. Given recent progress in molecular genetic methods, it is possible to obtain high-quality genetic information on genome properties by assembling chromosome-level genomes [11–13], analyzing genome-wide variation in the rate of mutation [14–16] and recombination [17–20], and measuring functional-genomic patterns of activity [21,22], and to then match these features with population-level whole-genome sequence data [23,24]. High-quality genomic data, population theory, and individualized null hypotheses from evolutionary simulations promise to be a powerful tool in population genetics to tease apart the genetic and evolutionary forces that govern genetic variation within and between species.

Variation in the mating system provides one crucial species-specific factor that influences traits, ecology, and population genetic parameters. For instance, self-fertilization as an extreme form of inbreeding acts to reduce the effective population size and the effective recombination rate [25–29], thereby leading to a reduction in heterozygosity, increased linkage disequilibrium, reduced influence of dominance, and increased variability in evolutionary trajectories due to the enhanced influence of drift and a concomitant reduction in efficiency of selection [30–37]. Within animals, the transition from outcrossing to selfing is often accompanied by accelerated reproductive incompatibility and isolation, relaxation of sexual selection and sexual conflict, degradation of mating ability, and the generation of outbreeding depression [38, 39]. In the context of population genomic analysis, it is the influence of self-fertilization on linkage disequilibrium and the way that it expands the genomic footprint of natural selection that is of particular interest. So the contrast between extreme linkage disequilibrium in self-fertilizing species and natural variability in recombination rate across the genome in outcrossing

species provides a unique opportunity to critically examine the interaction between population genetic and transmission genetic processes in shaping molecular variation within species.

*Caenorhabditis* nematodes are primarily outcrossing species with males and females [40], with the exception of three predominantly self-fertilizing hermaphroditic species: *C. elegans*, *C. briggsae*, and *C. tropicalis*. The self-fertilizing mode of reproduction appears to have evolved independently within each species and to have done so from an outcrossing ancestor fairly recently [41,42]. Sex in *Caenorhabditis* species is determined by sex chromosome dosage (X), with females and hermaphrodites having two copies of the X (XX) and males only one (X0) [43]. Because sex is determined by the absence of the X, males can arise spontaneously from nondisjunction of the sex chromosome during meiosis (~0.1% cases for *C. elegans* [44,45]), allowing rare outcrossing to occur in natural populations [46–49,50–52]. *C. elegans* is a model species for behavior genetics [53–56], development biology [57–59], and experimental evolution [45,60–62]. Moreover, the occurrence of hermaphroditism in *Caenorhabditis* nematodes can be attributed to changes in just a few pathways [63–66], while sex determination itself can be genetically manipulated, making *Caenorhabditis* species an unprecedented tool for study of the effects of reproduction mode in metazoans [45,67–70]. From a population genomic point of view, the genomic landscapes of diversity in all three selfing *Caenorhabditis* species are similar, with a consistent pattern among all chromosomes of higher genetic diversity on the peripheral "arm" regions and lower diversity on the central regions of chromosomes [49,50,71,72]. This pattern closely mirrors the chromosome-level pattern of recombination rate, which is high in chromosome arms and low in centers, with the latter occupying about half of the chromosome length [49,72–74]. To date, however, no comprehensive population genomic data are available for outcrossing *Caenorhabditis* species.

How might outcrossing be expected to affect the distribution of diversity across the genome? In this study, we use a previously constructed chromosome-level assembly [75] and a high-density genetic map generated here to examine whole-genome DNA sequence diversity for *Caenorhabditis remanei*. *C. remanei* is an obligate outcrossing nematode that has a significantly larger effective population size and levels of molecular polymorphism than selfing species [76–80]. To provide a consistent basis for comparison across species, we also reanalyzed a local sample of *C. elegans* from Hawaii (from [81]) to compare it with *C. remanei* using the same set of diversity statistics. In addition to inferring demographic histories, patterns of linkage disequilibrium, and genome-wide patterns of divergence, selection, and the spectrum of nucleotide substitutions, we performed evolutionary simulations under different evolutionary, mutational, and recombinational scenarios to compare patterns of diversity in *C. remanei* and *C. elegans* with theoretical expectations. We find that the mode of reproduction strongly determines overall levels of diversity and that finer chromosome-level differences in polymorphism are governed by the interaction of selection, mutation, and recombination, indicating that a comprehensive understanding of evolution, demography, and genetic transmission are needed to interpret whole-genome evolution.

## Results

### Recombination landscape of *C. remanei*

To generate the first genetic map for an outcrossing species of *Caenorhabditis*, *C. remanei*, we crossed two inbred lines (PX506 and PX553) derived from isolates collected near Toronto, Canada ([75], Fig 1 and S1 Table) and individually genotyped 341 F2 offspring. Of the 1,399,638 polymorphic sites among the parental strains, an average of 106,071 markers were covered by bestRAD-sequencing. Full filtration for informative markers yielded 7,512 total sites across the genome, which in turn were used to construct the genetic map. The total length

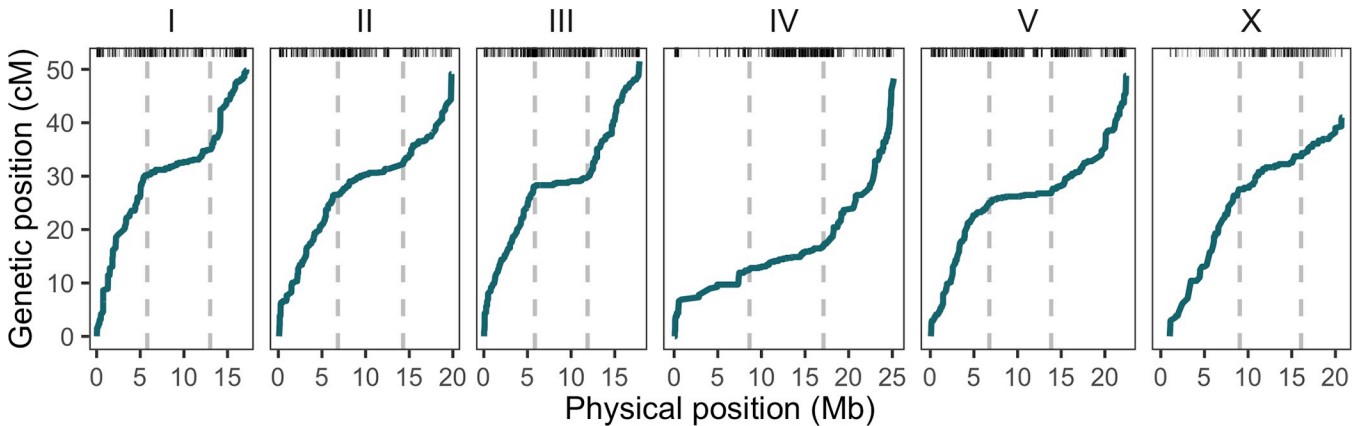

**Fig 1. Marey map of genome-wide patterns of *C. remanei* recombination.** Rugged marks on the top reflect the density of markers, while the dashed lines show the boundaries of the central domains (see Table 1). The recombination landscape along *C. remanei*'s holocentric chromosomes resembles that of other *Caenorhabditis* species, *C. elegans* and *C. briggsae*, with core domains ("centers") of low recombination and peripheral domains ("arms") of uniform and high recombination.

of the genetic map is 288.72 cM. Of this, 40.99 cM is the X chromosome determined via the female parent, and the remainder is the result of sex-averaged maps for each autosome (see details in S1 Table). The X chromosome has lower rates of recombination than the sex-averaged map of autosomes, which is likely driven by differential control of recombination on the sex chromosome (for example [82,83], reviewed in [84]), and/or by unequal recombination rates in males and females ([85–87], also reviewed in [88]). Our crossing approach precludes the construction of sex-specific maps, although examining any potential differences in recombination between males and females could be of interest to future studies.

For the purposes of the population genomic analysis, we are particularly interested in the shape of the recombination landscape across the whole genome. We find the chromosomal recombination landscape to be non-uniform in *C. remanei* in a fashion that is superficially similar to that seen in other *Caenorhabditis* species such as *C. elegans*, *C. briggsae*, and *C. tropicalis*, with ends of the *C. remanei* chromosomes having elevated recombination rates and for rates of recombination within a given chromosomal domain (arm or center region) being fairly uniform (Fig 1, [49,72–74,89]). Using stepwise regression to identify the boundaries of the central domains of lower recombination (Table 1), we find regions of low recombination

**Table 1. Positions of low recombination domains on chromosomes of the *C. remanei* genome obtained from crosses of the PX506 and PX553 strains.**

| Chromosome | Left arm ends, Kb (95% CI) | Right arm starts, Kb (95% CI) | Recombination rate, cM/Mb (left, central, and right domains) | Chromosome size, bp |
|---|---|---|---|---|
| I | 5,803 (5,753–5,853) | 13,007 (12,942–13,071) | 5.197, 0.673, 3.529 | 17,247,545 |
| II | 6,828 (6,779–6,878) | 14,292 (14,060–14,524) | 3.883, 0.808, 2.938 | 19,935,723 |
| III | 5,849 (5,828–5,870 | 11,888 (11,863–11,914) | 4.791, 0.316, 3.601 | 17,877,849 |
| IV | 8,639 (8397–8,881) | 17,108 (17,072–17,145) | 1.447, 0.520, 3.615 | 25,790,997 |
| V | 6,767 (6,717–6,817) | 13,852 (13,737–13,966) | 3.697, 0.270, 2.531 | 22,502,457 |
| X | 9,040 (8,960–9,120) | 16,070 (15,498–16,642) | 3.047, 0.879, 1.339 | 21,501,900 |

in *C. elegans* tend to be roughly one-third larger in relative size than in *C. remanei*. Specifically, the central domains of chromosomes I, II, III, IV, V, and X of *C. elegans* represent, correspondingly, 48%, 48%, 48%, 52%, 51%, and 36% of the total chromosome (estimated from Table 1 in [49]), while in *C. remanei* these represent 42%, 37%, 34%, 33%, 31%, and 33% of chromosomal length, which could be potentially caused by DNA loss in selfing nematodes and relative reduction of the arms [90].

## Genetic diversity of *C. elegans* and *C. remanei*

In order to compare the landscape of genomic diversity in the outcrossing *C. remanei* to those of primarily self-fertilizing species, we sequenced 14 diploid genomes of *C. remanei* individuals collected in a forest area near Toronto, Canada. To our knowledge, this work represents the first comprehensive analysis of chromosome-scale patterns of diversity for outcrossing species from *Caenorhabditis*. Consistent with patterns observed for selfing species of *Caenorhabditis*, nucleotide diversity is as much as 40% higher in the regions of high recombination than in the central domains on all chromosomes (*C. remanei* mean $\pi \pm$ SD in arms, centers, and total: 17 x $10^{-3} \pm 4.9$ x $10^{-3}$, 12 x $10^{-3} \pm 4.2$ x $10^{-3}$, and 15 x $10^{-3} \pm 5.2$ x $10^{-3}$, see Fig 2). Average levels of polymorphism agree qualitatively with previous estimates from this species based on the analysis of individual genes [79,80,91].

To directly compare equivalent samples of *C. remanei* and *C. elegans*, we reanalyzed 28 wild *C. elegans* isolates collected at a single location in Hawaii (data from [81], S1 Table) and calculated the diversity landscape across the genome using the same analysis pipeline that we applied to *C. remanei*. Diversity for *C. elegans* was assessed for individual diploid genotypes rather than isotypes, as was performed in the original study, so as to properly retain information regarding genotype frequencies within the population. Consistent with previous reports, this analysis show that *C. elegans*, like *C. remanei*, has higher diversity levels on the arms compared to the centers [50,81,92–94], with mean values across all site types ($\pi \pm$ SD) in arms, centers, and total of 1.8 x $10^{-3} \pm 2.4$ x $10^{-3}$, 0.54 x $10^{-3} \pm 1.2$ x $10^{-3}$, and 1.2 x $10^{-3} \pm 2.1$ x $10^{-3}$.

The patterns of nucleotide diversity in *C. remanei* are qualitatively similar to *C. elegans* in distribution across the genome. However, nucleotide diversity differs quantitatively from *C. elegans* in scale by being higher by roughly one order of magnitude, consistent with previous observations of substantial reduction of diversity in selfing *vs*. outcrossing species of *Caenorhabditis* [76–80,91,95,96]. Highlighting this point, the number of SNVs that we used in the analysis (after filtering, masking of repeats, regions with low mappability, indels, and their flanking regions) was 243,456 variants for the *C. elegans* sample and almost ten times more, 2,365,750, for *C. remanei*.

When comparing patterns of polymorphism between domains of high and low recombination, we find that, as measured by $\pi$, chromosome arms are significantly more diverse in general than chromosome centers for both species, which also holds when looking specifically within exons and introns. Within a given domain, we find that introns are much more diverse than exons within *C. remanei* but not significantly different in *C. elegans* (see Table 2). As described below, this difference across functional groups has undoubtedly been caused by reduced effective recombination rate in *C. elegans*, which has made these domains highly susceptible to selective sweeps and background selection, homogenizing diversity across linked genetic elements [79]. In addition, hyper-divergent haplotypes, located mostly in the regions of high recombination, contribute to the difference in diversity among domains [97]. Consistent with this idea, variance in $\pi$ is nearly twice as large in chromosome arms as in central domains within *C. elegans* but fairly similar across the chromosome in *C. remanei*. Genomic patterns of other diversity statistics such as θ, Tajima's D, variance, skew, kurtosis, the number

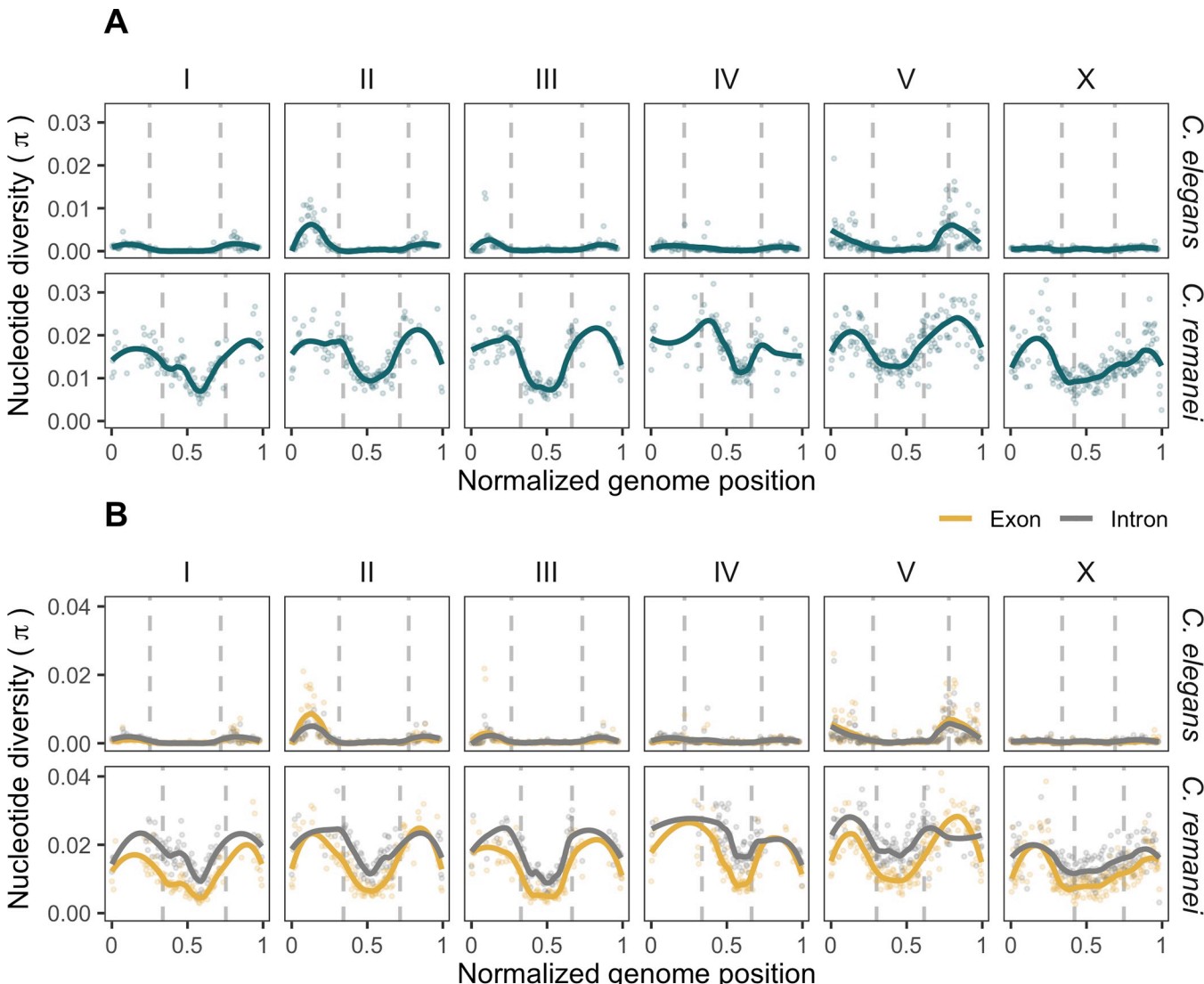

**Fig 2. Diversity landscapes of *C. elegans* and *C. remanei*.** Dots represent nucleotide diversity per 100 kb window, lines show locally weighted smoothing of these values, and the vertical dashed lines are the boundaries of regions of low recombination central domain for *C. elegans* from [49] and for *C. remanei* from this study. **(A)** Nucleotide diversity per 100 kb non-overlapping windows, windows with less than 10% of coverage were removed. Quantitatively, outcrossing *C. remanei* has one order of magnitude greater nucleotide diversity than partially selfing *C. elegans*. However, qualitatively, both species have significantly greater diversity in the regions of high recombination. **(B)** Nucleotide diversity within exons (yellow) and introns (grey) of protein-coding genes summarized per 100 kb non-overlapping windows, windows with less than 5% of coverage were removed. *C. remanei* shows a large and substantial difference in diversity between exons and introns along the genome, unlike *C. elegans*.

of haplotypes, H1, H12, H2.H1, ZnS, omega (see for details on statistics in [98], and β [99] are shown in S2 Fig).

## Divergence and the spectrum of substitutions in *C. remanei*

To facilitate examining the landscape of rates of divergence across the *C. remanei* genome, we reconstructed ancestral states for the *C. remanei* genome by using several genomes of *C. remanei* and a reference genome of *C. latens* [40,80,90]. In so doing, we imputed ancestral states for 56% of the PX506 reference genome (60%, 56%, 58%, 42%, 59%, 68% from the lengths of the chromosome I—X), which included 3,553,584 nucleotide substitutions; after filtration of

**Table 2. Statistical comparisons of nucleotide diversity ($\pi$) in *C. elegans* and *C. remanei* populations within different genomic domains.**

| Comparison | Species | Cohen's d | Permutation test, Z | Permutation test, *P*-value |
|---|---|---|---|---|
| $\pi$ within arms vs centers | *C. elegans* | 0.62 | 9.28 | $< 10^{-4}$ |
| | *C. remanei* | 1.08 | 13.96 | $< 10^{-4}$ |
| $\pi$ within exons on arms vs centers | *C. elegans* | 0.52 | 7.94 | $< 10^{-4}$ |
| | *C. remanei* | 1.36 | 16.16 | $< 10^{-4}$ |
| $\pi$ within introns on arms vs centers | *C. elegans* | 0.65 | 9.73 | $< 10^{-4}$ |
| | *C. remanei* | 0.70 | 8.33 | $< 10^{-4}$ |
| $\pi$ within exons vs introns on arms | *C. elegans* | 0.06 | 1.03 | 0.8456 |
| | *C. remanei* | 0.43 | -5.34 | $< 10^{-4}$ |
| $\pi$ within exons vs introns on centers | *C. elegans* | 0.04 | -0.56 | 0.2927 |
| | *C. remanei* | 1.36 | -15.86 | $< 10^{-4}$ |

genomic windows by coverage we used 2,657,650 substitutions, 53% of which were located on arms. All chromosomes had comparable fractions of substitutions from the ancestor (5 ± 0.24%); however, our inference may have been affected by the use of only one reference of *C. latens*, as these species are closely related and probably have some unresolved ancestral polymorphisms. Calculating the Tamura distance [100] of the PX506 reference genome from the reconstituted ancestral genome showed that overall divergence was 1.6 times greater on the arms than in the central domains (Fig 3A). Divergence tends to be positively correlated with recombination [101]. While the divergence landscape largely resembled the pattern of nucleotide diversity for the Toronto population of *C. remanei* examined here, chromosome regions showing low divergence tend to be wider than those seen for diversity measures in the regions of low recombination. The longer divergence time between *C. latens* and *C. remanei* has allowed for resolution of population genetic processes (fixation) that are ongoing within the Toronto population (segregating variation), which resulted in more pronounced differences between the domains of the divergence landscape compared to the diversity landscape, especially near domain boundaries (Fig 3A and 3C vs Fig 2). The fluctuations in the ratio of nucleotide diversity over divergence on the boundaries of recombination domains are especially noticeable on chromosomes I, II, III, and V (Fig 3B). The diversity/divergence ratio fluxes could result from selective processes, demography, or non-uniformity in the mutation landscape (S3A, S3B and S3D Fig). The divergence is noticeably lower in exons than in introns, 2.6 times less in the central domains and 2.7 in the arms, and significantly lower in domains of lower recombination (Exons: $d = 1.13$, $Z = 54.40$, *P*-value $< 10^{-4}$; Introns: $d = 1.63$, $Z = 69.26$, *P*-value $< 10^{-4}$; Total: $d = 0.28$, $Z = 13.55$, *P*-value $< 10^{-4}$) indicating strong selection.

Using the inferred ancestral states and the *C. remanei* reference genome (strain PX506) to estimate the rate of nucleotide substitution, we find that all types of substitutions show a consistent pattern across the chromosomes (S3B Fig). The transition over transversion bias with standard deviation estimated in 1 Mb windows in this comparison is 1.16±0.1, which is smaller than the 1.5±0.1 ratio observed within the *C. remanei* population sample from Toronto (Kruskal-Wallis $\chi^2 = 306.94$, *P*-value $< 10^{-16}$). The difference in biases may be attributed to shifts in the mutation spectrum [102] within the *C. remanei* population, as well as disparities in the mutation spectrum of *C. latens* used in the ancestral inference. Transition substitutions from C→T and G→A are the most common, consistent with previous observations of the *C. elegans* mutation patterns [97,103–106]. The genomic landscape of recombination also has an important effect on the nature of the substitutions, with more C→T and G→A substitutions in the central domain than in the arms, and more C→G and G→C in the arms (S3A Fig and

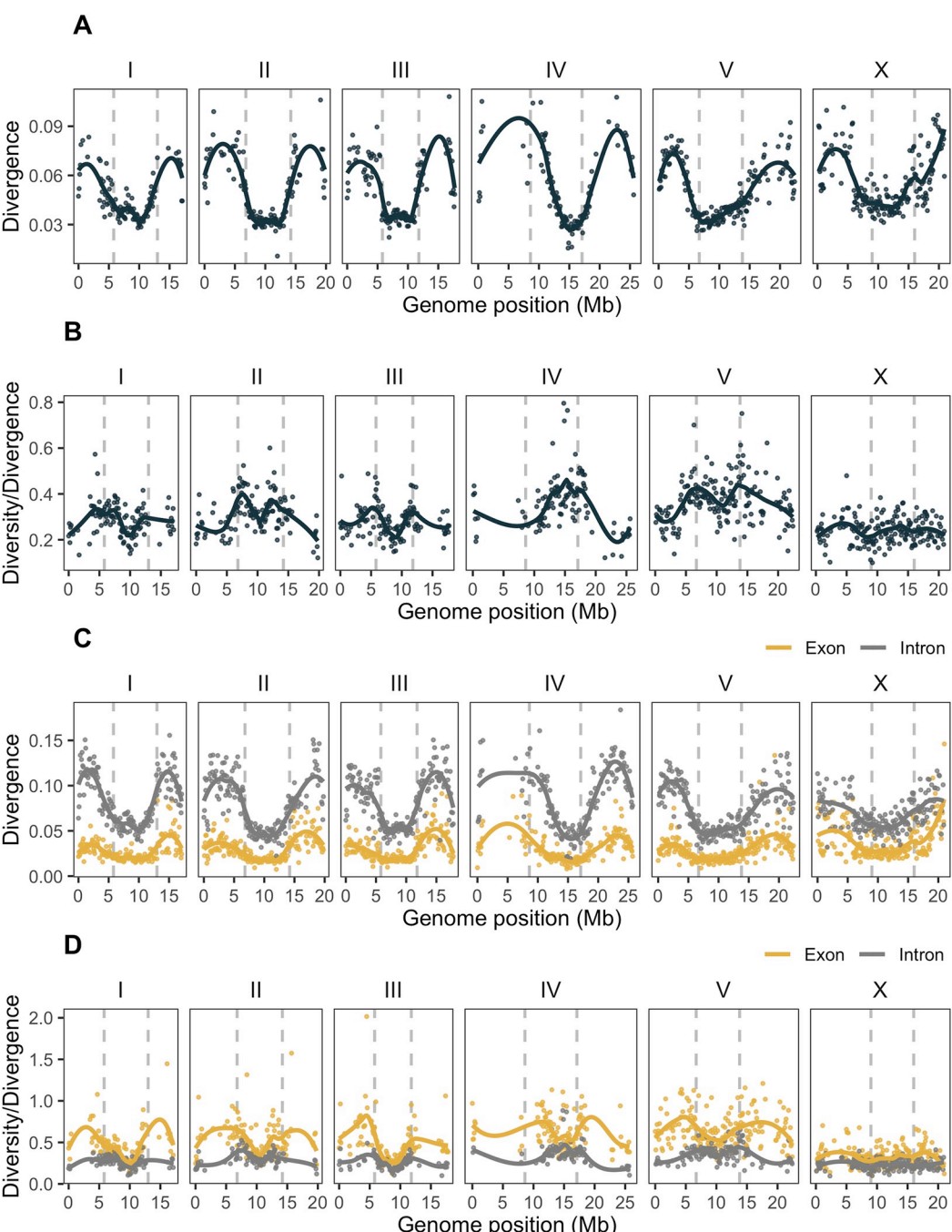

**Fig 3. Divergence of the *C. remanei* PX506 strain from ancestral states inferred from the *C. latens* genome.** Each point represents the Tamura 1992 distance on 100 kb non-overlapping filtered windows, with at least 30 kb of the window length containing ancestral states, with lines representing the locally weighted smoothing of these values. **(A)** Divergence of the reference genome (strain PX506) from the reconstructed ancestral genome. Divergence in central domains is 1.6 times lower than in the arms, and several highly divergent regions are located on the arms of chromosomes II, III, IV, and X. **(B)** Ratio of nucleotide diversity within the Toronto population of *C. remanei* over divergence shown above. **(C)** Divergence of the reference *C. remanei* genome from the ancestral states in exons (yellow) and introns (grey). Only windows with more than 5 kb of ancestral states within exons or introns are shown. Divergence in exons and introns is, respectively, 1.7 and 1.6 times less in the central domains than in the arms. **(D)** The ratio of nucleotide diversity in the Toronto population of *C. remanei* over the divergence estimated in exons (yellow) and introns (grey).

S2 Table). In the *C. remanei* population, the distribution of segregating substitutions along the genome follows comparable patterns to the inferred ancestral patterns (S3B and S3C Fig).

Genome-scale variation in the nucleotide substitution pattern within *C. remanei* could be a consequence of the interaction of recombination, selection, and demographic processes but could also be generated by variation in the mutation process itself. Experiments with mutation accumulation lines in *C. elegans* have shown a 1.2–1.6 times higher rate of base substitutions on the arms relative to the centers of chromosomes [103], probably due to mutagenic effects of recombination through double-strand breaks [107–109]. Other differences between these genomic regions also might contribute, such as chromatin organization, gene density and their expression, and transposon activity. Accurately inferring the mutation landscape of *C. remanei* will require further research, including the use of mutation accumulation lines and/or extensive sequencing of parent-offspring trios.

### Population structure, outcrossing, inbreeding, and effective recombination

Sampling nematodes from a single narrow geographic location can help to describe fine-scale population organization. Most of the nematodes in the *C. remanei* sample formed one cluster of related individuals, with a few more genetically distant nematodes (S4 Fig). Unexpectedly, the cluster was most likely formed from individuals from a single-family lineage displaying intensive inbreeding. This view is supported by relatively high $F_{is}$ values along the *C. remanei* genome (0.38 ± 0.15, S2M Fig). Such a value can be obtained through just a few generations of sibling mating (see Table 5.1 in [110]), which brings new insights into the biology of *C. remanei*. The inbreeding coefficient is even higher in the *C. elegans* population (0.86 ± 0.22), as expected for selfing or partially selfing species. This $F_{is}$ value corresponds to a 7.5% outcrossing rate ([111]; 1-s, where $s = 2F_{is}/(1 + F_{is})$ under the assumption of the equilibrium), falling within the range of values estimated for other *C. elegans* samples (Table 3). The effective outcrossing rate estimated from interchromosomal linkage disequilibrium (LD) is three orders of magnitude lower, 0.002% ± 0.005% (Table 3). This *C. elegans* sample consisted primarily of individuals derived from several distinct genetic lineages, which have been combined into "isotypes" in [50] and [81] and the CeNDR database (https://www.elegansvariation.org, S4 Fig).

Patterns of genome-wide linkage disequilibrium are drastically different in the *C. elegans* and *C. remanei* samples. *C. elegans* has very large blocks of LD both within and across chromosomes, with LD decaying slowly along the entire length of the chromosome (Figs 4 and S5), in agreement with theory [35] and previously reported results [71]. The degree of LD also varies significantly across chromosomes in *C. elegans* ($d = 0.58$, $Z = -151.1$, $P$-value $< 10^{-4}$), which is also consistent with previous observations [46,47]. In contrast, LD within the *C. remanei* population decays rapidly, within a few hundred base pairs on autosomes and somewhat more

**Table 3. Outcrossing rate of *C. elegans* samples reported in different studies.** The Method column specifies the approach used to estimate the outcrossing rate.

| Method | Outcrossing rate (%) | Study |
|---|---|---|
| Heterozygosity | 1.3 | [93] |
| | 20 | [112] |
| | 1.7 | [46] |
| | 7.5 | This study |
| Linkage disequilibrium | 0.013, 0.005 | [93] |
| | 0.0016 to 0.22 | [47] |
| | < 0.011 | [71] |
| | 0.0024 | This study |

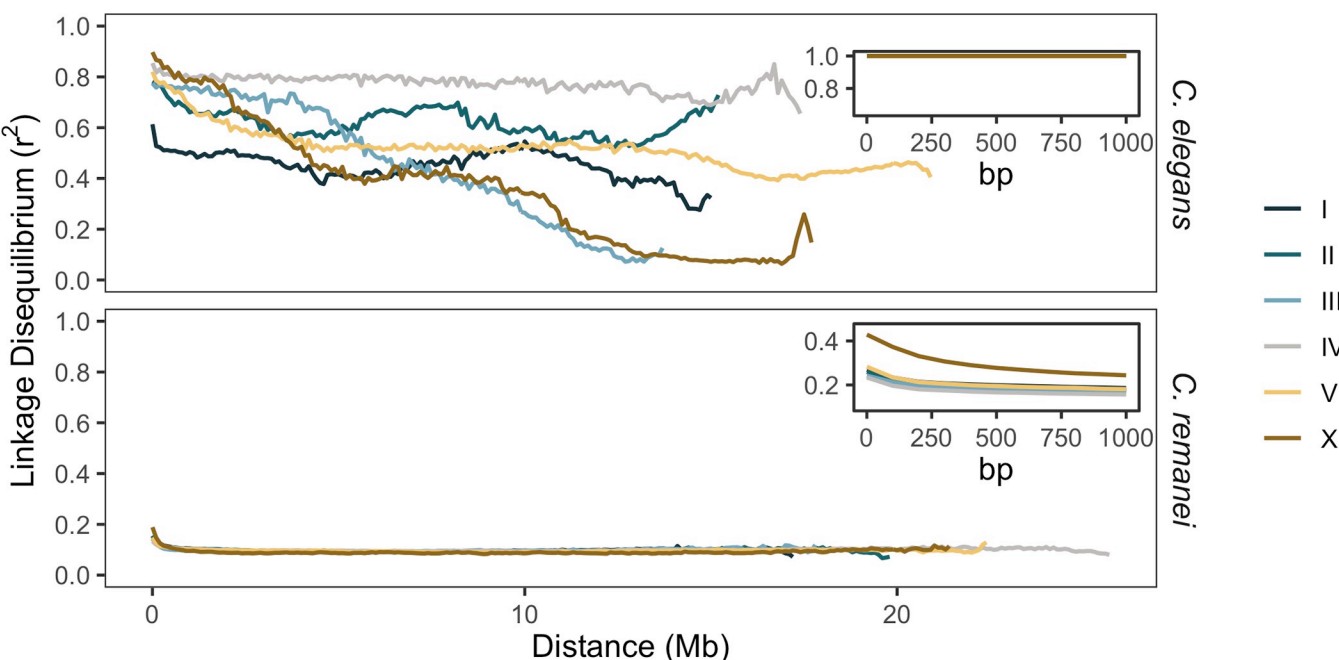

**Fig 4. The decay of linkage disequilibrium (LD) along the chromosomes of *C. elegans* and *C. remanei* populations.** The median values were estimated from LD between all biallelic polymorphic sites in *C. elegans* and every tenth site for *C. remanei* within either 100 kb non-overlapping windows (overall plots) or 100 bp windows (inset plots). Each line represents a chromosome.

gradually on the X chromosome (Fig 4), consistent with previous localized observations for this species [78]. Background inter- and intrachromosomal LD is estimated to be 0.1, which is more than a null expectation (the inverse of the sample size, 0.04) and probably related to inbreeding or another demographic process in the collected sample (S5 Fig). Consistent with these observations, the inferred genome-wide effective recombination rate in *C. elegans* is on average 8.4 times lower than in *C. remanei* ($3.8 \times 10^{-4} \pm 2.7 \times 10^{-4}$ versus $32 \times 10^{-4} \pm 77 \times 10^{-4}$; S6 Fig). However, the effective recombination in *C. elegans* does not follow the recombination domain structure. Here, the difference in meiotic and effective recombination rates among species is clearly driven by the increase in linkage disequilibrium caused by self-fertilization.

## Demographic history of *C. remanei* and *C. elegans* samples

Inference of demographic dynamics within both species reveals dynamic changes in population size over time (Fig 5). The first striking difference is that the estimated effective population size of our *C. remanei* sample is approximately two orders of magnitude higher than that of *C. elegans*, spanning a period of thousands of generations (Table 4). The effective size of the *C. elegans* sample from Hawaii has changed dramatically in recent generations, likely because of its intricate metapopulation structure [93,112]. From a historical point of view, the *C. elegans* sample displays a pattern of a precipitous decline in the effective population size toward the present, as previously noted [71]. In contrast, the Toronto population of *C. remanei* studied here maintained a consistent, relatively large effective population size but also displays notable fluctuations in size over time (Fig 5).

To look for site-specific changes in diversity that might be indicative of the action of natural selection, we also inferred population history for the *C. remanei* population using reconstructed ancestral states and a framework for demographic inference (Relate, [114]). Overall, the pattern of demographic history using this approach is very similar to that reported above,

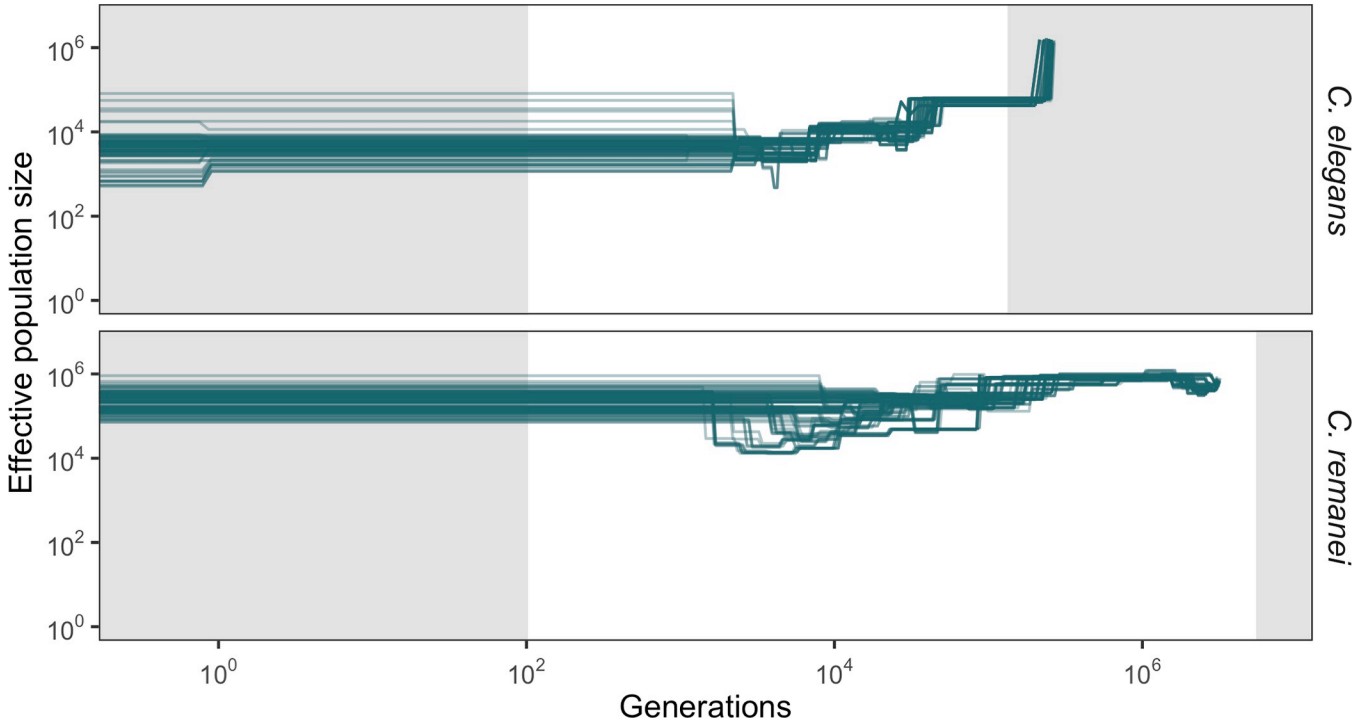

**Fig 5. Reconstructed demographic history of *C. elegans* and *C. remanei* populations.** Ancient demographic history based on SFS and LD information [113]. Calculations are based on 100 bootstrapped replicates using eight individuals from each species, with each line representing one replicate. The grey shading on the left-hand side of the plot indicates the regions of recent demographic history where estimations using this method may be less reliable. The grey shaded areas on the right-hand side of the plot show the regions where the inference should also be less accurate as we move further back in time beyond the threshold of ~4Ne generations. In this analysis, we used one generation per year, and scaling of the mutation rate (x0.5) and coalescent time (x2) for *C. elegans*. Demography derived from individual chromosomes is depicted in S7 Fig. Population sizes of both species fluctuate significantly over time, which is likely to influence estimates of the coalescent population size. Consistent with this, coalescent population size estimates tend to be one order of magnitude less than the population sizes calculated from polymorphism data (Table 4). The long-term population size of *C. remanei* sample is around two orders of magnitude larger than those of *C. elegans*.

as well as being concordant across all six chromosomes (see S7 and S8A Figs). However, after adjusting for multiple comparisons, these estimates do not indicate any genomic region that had been subjected to substantial positive selection in the past (S8C Fig). Nevertheless, estimated p-values deviated from the expectation of a uniform distribution in a fashion that is more pronounced on the arms than on the centers, a possible signal of non-neutral processes

**Table 4. Contemporary effective population size (N_e) for *C. elegans* and *C. remanei* samples from different studies.** The Method column indicates the approach used to estimate $N_e$. For our study, the 95% confidence intervals for the mean $N_e$ are shown in parentheses.

| Species | $N_e$ x $10^3$ | Method | Study |
|---|---|---|---|
| ***C. elegans*** | 0.2–9.5 | Allele frequencies | [93] |
| | 0.1–10 | Allele frequencies | [92] |
| | 80 | Allele frequencies | [47] |
| | 0.01–10 | Allele frequencies | [112] |
| | 6.5 (4.2, 8.3) | Coalescent | This study |
| | 34 (31, 37) | Allele frequencies | This study |
| ***C. remanei*** | 1600 | Allele frequencies | [78] |
| | 250 (220, 280) | Coalescent | This study |
| | 1400 (1400, 1500) | Allele frequencies | This study |

(S8D Fig). This also is consistent with significant and large differences across diversity and divergence in exons vs. introns in *C. remanei* caused by selection, as discussed above (Figs 2B and 3C). We attempted a similar analysis for *C. elegans* but could not reconstruct a sufficient number of ancestral sites from several strains of *C. elegans* and its closest relative *C. inopinata* to allow the analysis to proceed.

## Evolutionary simulations

The empirical data reveal two major features. First, that total polymorphism is lower in the partial-selfing *C. elegans* relative to the outcrossing *C. remanei*, and second, that the genomic landscape of genetic polymorphism is structured and appears to be strongly correlated with domains of high and low recombination. Because many interacting factors can potentially influence these observations, we conducted individual-based simulations to better understand the separate and combined effects of positive selection, background selection, recombination, variation in mutation rates, variation in demography, and variation in rates of partial selfing on population genomic signatures within these species. We looked at three separate scenarios on a wide array of population genetic statistics using evolutionary simulations in SLiM [9]: the interaction of selfing rate, selection, mutation landscape, and recombination domains; the decay of the ancestral diversity; the effects of fluctuations in population size.

As predicted by theory [25–28,37,115], selfing dramatically reduces the diversity in the population, especially when combined with either positive or negative selection (Fig 6A, far right panels). Because recombination has little impact on within-lineage diversity under self-fertilization, any form of selection tends to eliminate the variation at linked sites, often at the scale of the whole genome. As a corollary of this, variation in recombination rate across the genome has little influence on the genomic landscape of polymorphism when the selfing rate is high because the genetically effective recombination rate becomes very low in that situation. Similarly, the expected variance in evolutionary outcomes is also very small when selfing and selection combine because selection consistently eliminates variation irrespective of when and where new mutations arise within the genome (Fig 6B).

When even minor amounts of outcrossing enter the population [118–121], however, the situation changes dramatically. In the neutral case, the genomic landscape of polymorphism remains flat regardless of outcrossing rate, as predicted (Fig 6A, [27,29,35]). But when selection is introduced, regardless of whether it is positive, negative, or balancing, then regions of high recombination maintain substantially more variation than regions of low recombination (Fig 6A, left most panels). These simulations, in particular, do a very good job of recapitulating the empirical patterns seen in both *C. elegans* and *C. remanei*. Recent inbreeding, as observed in our *C. remanei* samples, does not strongly disrupt this pattern, although the influence of recombination on the genomic distribution in polymorphism is reduced under balancing selection in face of inbreeding, as might be expected. Importantly, however, balancing selection does not generally lead to qualitatively different genomic patterns of polymorphism, nor does it change expectations of the influence of selfing on regions of high and low recombination (see also S10 and S11 Figs). The variance in outcomes among simulations also displays a recombination-dependent pattern (Fig 6B). Specifically, neutral scenarios have greater variance in regions of lower recombination [117]. Whereas the variance of π in those domains in non-neutral simulations is lower than in neutral ones and extremely low in scenarios with positive selection.

Although variation in recombination does an excellent job of capturing the genomic differences in polymorphism observed in our empirical examples, it is also possible that these patterns could be caused by domain-specific variation in mutation rate. Indeed, there is evidence

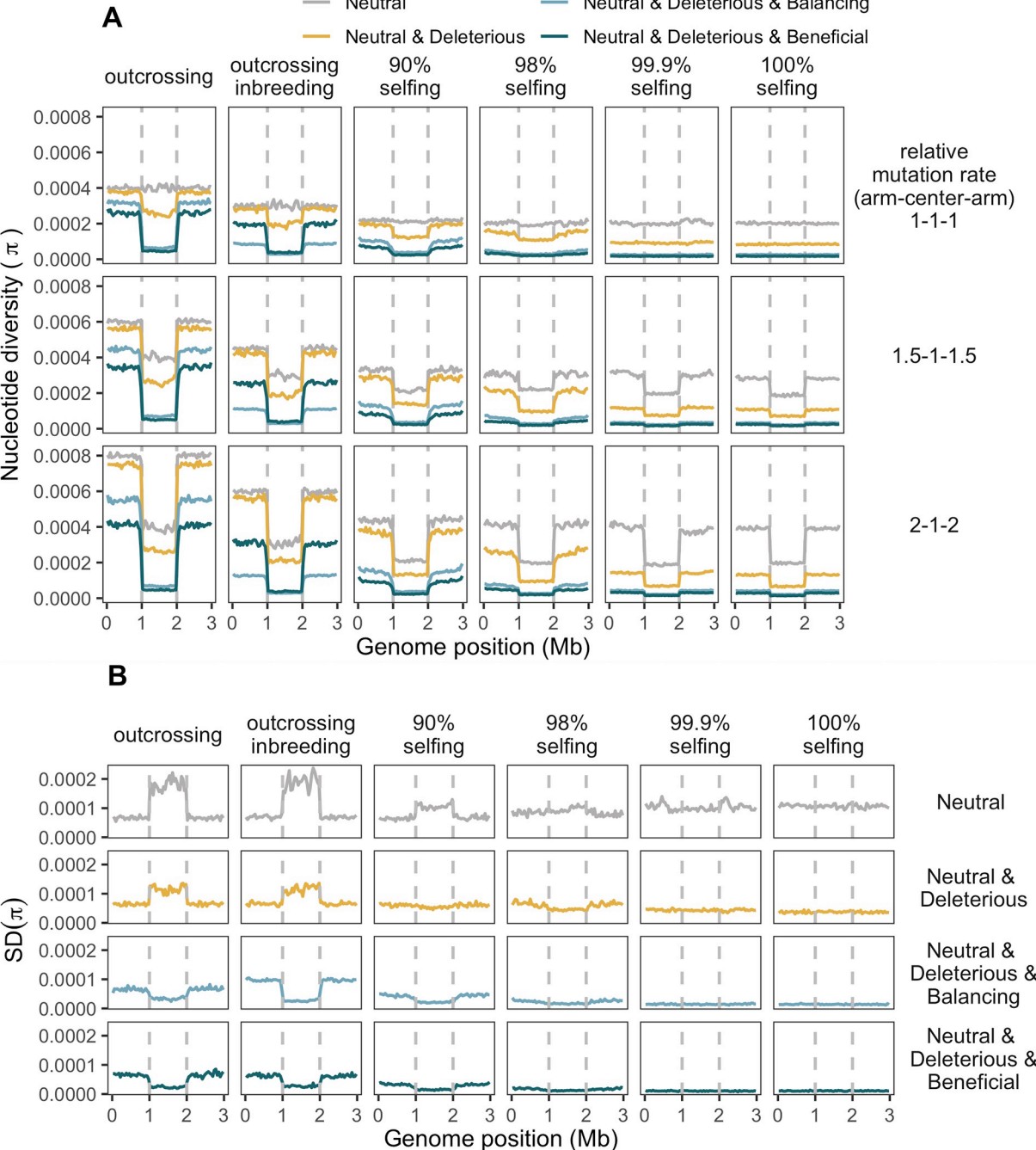

**Fig 6. Nucleotide diversity landscapes in simulated populations.** Lines represent nucleotide diversity per 40 kb window, the vertical dashed lines are the boundaries of regions of low recombination central domain. Columns show the outcrossing rate, where "outcrossing" means completely outcrossing populations and, in other columns, % specify the percentage of selfing in population; "outcrossing inbreeding" corresponds to scenarios with outcrossing populations that underwent the bottleneck at the very end of simulations (see Methods). Rows represent domain-specific differences in mutation rate, with 1-1-1 as the uniform mutation landscape, 1.5-1-1.5 as 50% more mutations in domains of high recombination, and 2-1-2 meaning two times more mutations in domains of high recombination. Colors show the selection regime (see Methods). **(A)** Mean nucleotide diversity per 40 kb non-overlapping windows. The mean values are lower in regions of low recombination in scenarios with selection and a non-uniform mutation landscape. Higher selfing rate and selection pressure reduce the coalescent time and, consequently, nucleotide diversity. **(B)** The standard deviation of nucleotide diversity in scenarios with uniform mutation landscapes. The variance in diversity gets lower with selection and reduction of effective recombination in non-neutral scenarios and becomes higher with increased selfing in the case of neutrality [116,117].

that mutation rates are different among recombination domains in *C. elegans* [103,106]. If the mutation rate is, in fact, elevated on the chromosome arms, then polymorphism also increases on those arms, with the difference between arms and centers increasing with the disparity in mutation rate (see Fig 6). As long as there is natural selection and a sufficient level of outcrossing, then changing the mutational landscape does not qualitatively alter expectations of the pattern of genomic variation with respect to the influence of domains of high and low recombination. The situation is more complex under neutrality and/or with a high degree of selfing, in which case the disparity in mutation rate can mimic the pattern of polymorphism expected under the combination of selection and recombination. Thus, while the pattern of variation in *C. elegans* appears to be consistent with some level of outcrossing combined with selection, it is also consistent with selfing and variation in mutation rate correlated with recombination (Fig 6). As outlined below, distinguishing these cases, therefore, depends on the combined effects of all of these factors on additional haplotype and site-based statistics, such as theta, the number of haplotypes, variance, kurtosis, H12, H1.H1, H1, β-statistics, $F_{is}$, omega, that are affected by the mutation landscape and other statistics, such as Tajima's D and LD based statistics (ZnS), that tend not to be (S10, S11 and S12 Figs).

Another formal possibility for the genomic pattern observed within *C. elegans* is that, while we might expect this species to exhibit a pattern consistent with selfing, there might be residual, *C. remanei*-like unresolved ancestral variation on the arms due to its transition from outcrossing to selfing (see [97]). To examine this possibility, we used evolutionary simulations to explore the decay rate of ancestral polymorphism from an outcrossing ancestor to a population experiencing either 98% or 100% selfing. For complete selfing and purely neutral variation, the ancestral pattern of variation does indeed persist even after $6N_e$ generations (Fig 7). However, the addition of any form of natural selection and allowing for a small fraction of outcrossing individuals leads to the rapid decay of ancestral diversity, within $1N_e$ generations for 100% selfing and $2N_e$ for 98% selfing. These results are consistent with the expected average coalescent time for the scenario (~1/N, more specifically (1+F)/2N, where F is the selfing rate; [25–27,29,35,122]). Here, the variance in the nucleotide diversity is considerably higher in the populations that transition to obligate selfing, which also agrees with previous studies [29,123].

The above simulations assume populations of constant size, but we know that variable sizes are the reality for most populations, especially those that might experience more of a "boom and bust" life cycle, such as that experienced by these species [124]. We, therefore, also conducted a third type of simulation that examined two types of recent change in the population size under neutral scenarios: 1) change from 15K to 5K every five generations for 100 generations after the burn-in; 2) exponential growth by 3% every generation for 100 generations. We then evaluated the diversity statistics and compared them for each simulation after 100 generations of changes in the population size and at the burn-in. S13 Fig illustrates the fold change in a broad range of population genetic statistics in neutral simulations, with large effects in different directions in many of the statistics used in the study, but especially notable in Tajima's D, haplotype-based statistics, skew, and kurtosis. Importantly, these statistics are sensitive to perturbations in population sizes, which occur naturally in wild nematode populations. Likely, the impact of recent demographic shifts on diversity might be less pronounced in the regions of low recombination with selection, as the variance in those scenarios is lower (e.g., Fig 6B). Thus, understanding population structure and local dynamics in a population is particularly valuable for interpreting observed patterns of diversity.

We attempted to distinguish the complex interactions among these evolutionary factors by training a convolutional neural network on diversity statistics and classifying population characteristics such as selfing rate, mutation landscape, and selection. The network performed well in simulations, but had difficulty classifying empirical samples of *C. elegans* and *C. remanei*

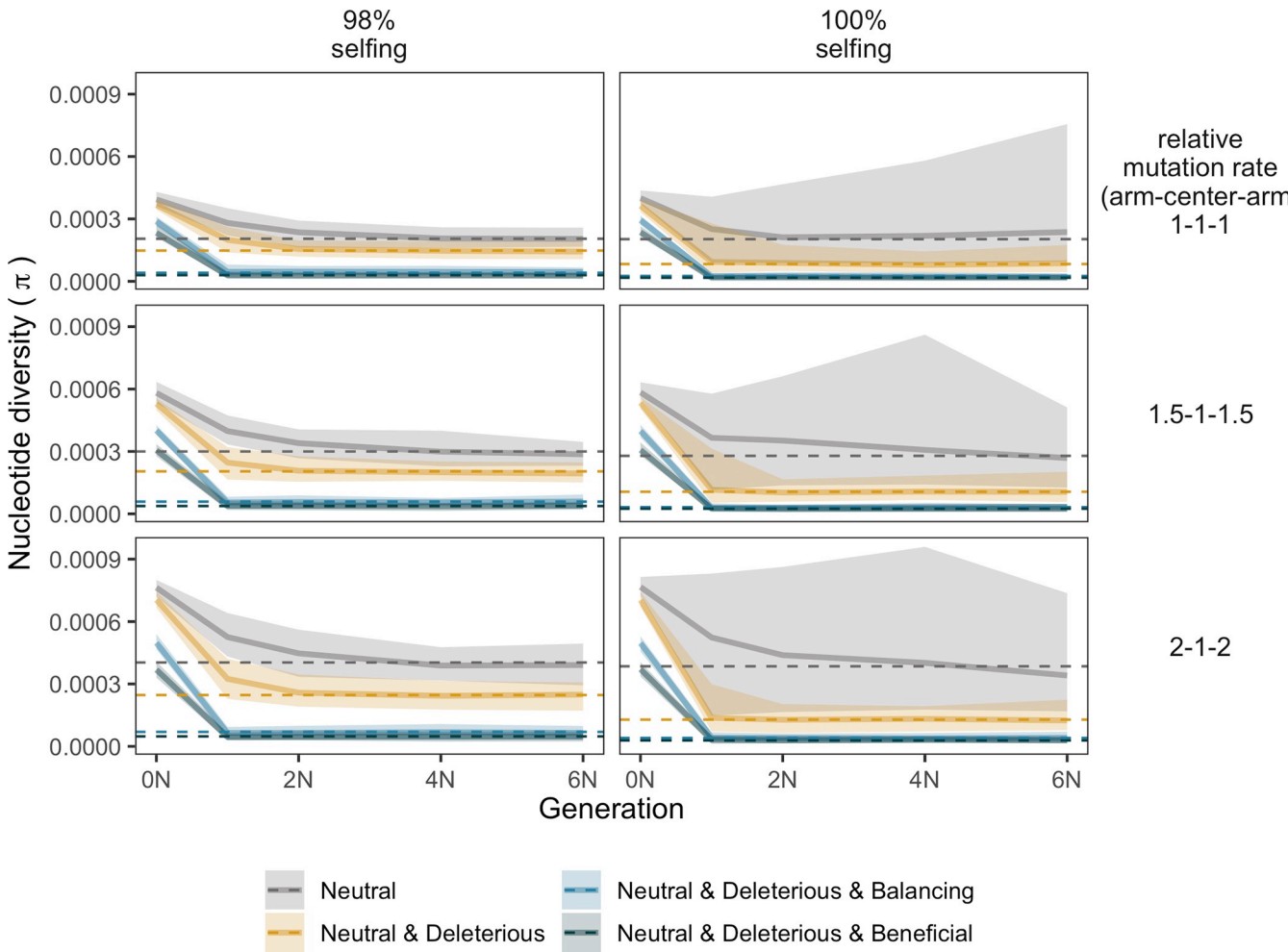

**Fig 7. Decay of ancestral polymorphism in simulated populations that transitioned from outcrossing to selfing.** The columns indicate populations that switched to 98% selfing (left column) and 100% selfing (right column) after obligate outcrossing. Lines represent the mean nucleotide diversity values estimated from 50 replicates in domains of high recombination, with the shaded areas representing the standard deviation among these replicates. Generations on the ordinate axis show the number of generations after the burn-in. Colors show the selection regime. Each row represents the mutation landscape (see Methods), and the dashed lines represent the nucleotide diversity in the populations without outcrossing ancestors. The average diversity in all scenarios with selection decline rapidly (within ~N generations) towards levels of diversity in selfing or partially selfing populations. Neutral scenarios in obligate selfing populations have greater variance, which increases over time because these populations are, basically, composed of independent lineages.

populations using this approach. This likely occurred because the successful application of this approach requires additional bioinformatic data, empirical information, and an expanded simulation framework that includes other factors such as demography (see S1 Data Appendix A1).

## Discussion

The transition from traditional population genetics to molecular population genetics shifted the general analytical framework from alleles at a locus to the nucleotide sequence at a particular site in the genome. The progressive expansion of this framework requires scaling these approaches in the context of broader genome-wide factors such as linkage, recombination, and localized variation in mutation rates, as well as how the impacts of these processes are amplified by population history and structure, and species biology. Attempting to understand the separate and combined impacts of these factors requires comprehensive information about

molecular diversity across the genome and a theoretical context in which different alternatives can be rigorously tested. *Caenorhabditis* nematodes provide a natural experiment in which phylogenetically close species have drastically different lifestyles, demography, and genetic processes. With this in mind, we assessed the genomic landscape of various population diversity statistics of populations of *C. remanei* and *C. elegans*. We find that the *level* of diversity is dramatically different across species, with partially selfing *C. elegans* having order of magnitude lower diversity than the outcrossing *C. remanei*, yet that the *pattern* of genetic diversity is strikingly similar across both species' genomes, being positively correlated with large-scale transitions in recombination rate between chromosome centers and arms.

## Mating systems and global patterns of diversity

Focusing first on global differences among species, here we used local samples of individuals to facilitate analysis of population history and structure and to allow comparison of features of diversity distributions among outcrossing and selfing species with similar genome organizations. Previous analyses of diversity of *C. elegans* at a global scale [81,97] have demonstrated that even with an overall low genetic diversity expected for a nearly selfing species, a large number of hyper-divergent haplotypes—covering one-fifth of the genome and located mostly on the chromosome arms—have been maintained in that species, possibly because they are involved in adaptation to specific environmental niches. These regions also could result from allelic sequence divergence, analogous to the Meselson effect [125] demonstrated for asexual species, expected from coalescent theory when the fraction of outcrossing is considerably lower than the reciprocal of the population size [118,119,122,126]. Overall, our estimates of the outcrossing rate and population size for the *C. elegans* sample overlap with previously reported values (Tables 3 and 4). It remains an open question if such hyper-divergent regions are present in populations of outcrossing *Caenorhabditis* species, because short-read data, such as that used here for *C. remanei* tends to be difficult to align in hyper-divergent regions of high recombination rate. This can potentially lead to lower diversity estimates, as regions with poor coverage are excluded from the analysis. Long-read data will be required to close these gaps in the genomic coverage.

Looking broadly at haplotype structure across the genome, the rate of decay in linkage disequilibrium (LD) was drastically different in *C. remanei* and *C. elegans*. As predicted by theory [35,127], the outcrossing *C. remanei* displays rapid LD decay, within hundreds of base pairs, and low levels of inter and intrachromosomal correlations. These observations and our estimations of genome-wide effective recombination rate are consistent with previous studies [78]. However, slightly elevated background LD could indicate a recent inbreeding, consistent with high $F_{is}$ values (S2M Fig). The "Scottish tartan" pattern of genomic linkage disequilibrium in *C. elegans*, in which associations across chromosomes are often nearly as strong as associations within chromosomes (S5 Fig), is particularly striking and is consistent with the previous observations of high inter- and intrachromosomal linkage disequilibrium in *C. elegans* [78]. Because effective recombination rate is a major driver of diversity and linkage along the genome, nematodes provide an especially useful model for studying the evolutionary effects of this factor.

## Genomic landscapes of diversity and factors that affect them in *C. elegans* and *C. remanei* populations

Moving from a global genome perspective to patterns of diversity observed along specific chromosomes requires teasing apart a large number of potentially influential factors (including population history, structure, and demography, natural selection, species biology and

development, and reproduction mode, location-specific variation in recombination and mutation, and additional genomic properties such as genome activity and positions of various genetic elements like transposons, genes, regulatory elements, etc.). Using the empirical data from populations of *C. elegans* and *C. remanei* and evolutionary simulations derived here, we step through below how some of these factors contribute to the landscape of diversity.

### Recombination landscape and genomic organization

We constructed the first comprehensive genetic map of *C. remanei*, the first for an outcrossing species in this genus (Fig 2), to guide the understanding of genomic patterns of diversity. The genetic map shows a similar structure to other maps constructed for *Caenorhabditis* species [49,73,72] with large central parts of chromosomes of suppressed recombination [49,73,89]. The recombinational landscape deduced from the population data was also consistent with this pattern. Notably, the X chromosome exhibits the same recombination pattern as autosomes in *C. remanei*, as in other *Caenorhabditis* species [49,73,89], but with a noticeably shorter genetic map. More research is needed to determine if this is due to distinct recombination regulation on the sex chromosome [82,88,128,129] or to averaging of recombination rates between sexes, which might have different recombination regulation, as has been shown for *C. elegans* [88,130]. *C. elegans* and *C. remanei*, and some other *Caenorhabditis* species, share other similarities in chromosome organization: central parts of chromosomes display lower recombination and have higher gene density, lower repetitive content, lower GC-content, higher gene expression, and a higher level of inter-chromosomal interactions than the peripheral parts of chromosomes [75,131,132]. As recombination across the genome is one of the critical factors that might affect the shape of diversity landscape, and the genetic map is necessary to model and interpret diversity along the genome, we used the domain-like structure of *Caenorhabditis* chromosomes as the basis for our simulations to allow comparison of diversity landscapes between domains in empirical and simulated populations.

### Diversity landscapes and linkage-disequilibrium

Self-reproduction leads to increased homozygosity and linkage disequilibrium, which is known from theory [25–27,29,122,133–135] and which has been frequently shown empirically (reviewed in [136]). For *Caenorhabditis* nematodes, genomic landscapes of diversity have already been derived for three selfing species [50,71,72]. In order to compare the diversity patterns of nematodes with different reproduction modes, we obtained the first genome-wide diversity landscape for an outcrossing *Caenorhabditis* species, *C. remanei*, and contrasted its patterns with the selfing *C. elegans*. *C. remanei* has one order of magnitude higher nucleotide diversity than *C. elegans*, due to larger population size, higher effective recombination, and outcrossing, which is consistent with previous gene-based estimates [76,91]. Importantly, the diversity landscapes across the genome in both species follows the domain-like organization of chromosomes with higher diversity levels in the peripheral regions of higher recombination and low diversity in the "central" domains of lower recombination of their holocentric chromosomes.

In the absence of selection, our simulations show that nonuniform recombination rates by themselves cannot generate the structure of diversity observed in the empirical data, although recombination rate did, of course, profoundly impact the majority of population genetic statistics (Figs 6 and S10, neutral scenarios in the top rows compared to Figs 4 and S2). In general, these patterns agree with previous predictions expected for a number of these statistics (see Table 1 in [37]).

## Interaction of selection and recombination

Genetic variation and recombination rate tend to be correlated due to the reduction of diversity of linked sites both by hitchhiking (positive) and background (negative) selection [4,30,32,137–140]. Positive correlations of meiotic recombination rate and nucleotide diversity have been shown for many species [108,141–148], however, this pattern is not universal and can be influenced by a variety of factors [107,149]. Selfing generates a distinct reduction in effective recombination, thereby reducing the efficiency of selection [26,33,36,122,150,151]. Both *C. elegans* and *C. remanei* display higher diversity in regions of higher recombination, which may be a signature of linked selection. This is consistent with prior research showing that *Caenorhabditis* nematodes exhibit substantial effects of background [143,146,152,153], positive selection [50], and perhaps balancing selection on some loci [97,154]. Our observation that exons and introns show very strong differences in diversity in *C. remanei* whereas they are very similar in *C. elegans*, provides strong evidence that selection is much more efficient in *C. remanei* than *C. elegans*, almost certainly because reduced effective recombination rates in *C. elegans*.

The genome organization across *Caenorhabditis* as a whole appears to be likely the result of prolonged continuous selection, as chromosomes within all species are compact, densely packed with genes, and have similar patterns of recombination, and chromatin activity [75]. They also tend to lack repetitive elements in the regions of lower recombination, which are likely to have been removed by background selection [132]. Similarly, the central gene-dense regions in domains of lower recombination also tend to be more conserved. For example, within *C. remanei*, reconstructing ancestral states using *C. latens* genome as an outgroup reveals that the central chromosome domains are twice as conserved as the arms, a pattern also observed in *C. elegans* [131].

But exactly how do recombination and selection interact to generate these patterns? To address this question, we performed evolutionary simulations using the *Caenorhabditis* domain-like recombination landscape under conditions of different selfing rates, patterns of deleterious and beneficial mutations, as well as mutations under balancing selection. By including selection, we were able to mimic the shape of some of the diversity statistics found in empirical data (Figs 2, S2, S6, S10, S11 and S12). Importantly for the interpretation of diversity within *C. elegans*, even a small proportion of outcrossing ($>1/N$) was enough to start observing the effects of linked selection, as is predicted by theory [120–122]. So, at first glance, it would appear that we can do a very good job predicting patterns of genomic diversity within these species using the right balance of natural selection integrated across domains that differ dramatically in recombination structure. However, diversity is not shaped solely by the forces that either restructure or remove it from the population, but also by mutational forces that introduce it into the population in the first place.

## Mutation landscapes

Mutation is the initial source of genetic variation, and the mutation rate can fluctuate along the chromosome due to such factors as chromatin accessibility, methylation, transcription activity, recombination rate, genomic context, replication and reparation timing [155–164]. Nonuniform mutation landscapes have been observed in various species [165–168], including *C. elegans* [103,106], where the mutation rate on the arms is 1.2–1.6 higher than in the central regions. For these nematodes, these differences might be caused by the mutagenic properties of recombination itself, which varies strongly across the genome, or by a variety of other potential factors. To explore potential variation in mutation rates across the genome, we inferred the substitution spectrum for *C. remanei* using our inference of ancestral genomic

states. This spectrum has the same dominant type of substitution, C→T|G→A, as *C. elegans* [103,169,170], and these are the most common polymorphic types of biallelic substitution both in *C. elegans* and *C. remanei* samples (resp. 58.4% and 60% from all types of biallelic substitutions). The fraction of C→T|G→A, A→T|T→A, and C→G|G→C substitutions are different within regions of higher and lower recombination in *C. remanei* (S3 Fig). The similarity of both chromosome organization and substitution spectrum of *C. remanei* and *C. elegans* implies that the mutation rate in *C. remanei* might be greater in regions of high recombination. However, direct mutation accumulation experiments and more detailed analysis of mutational signatures and subtypes are required to fully describe the features of the mutation landscape of *C. remanei*. Similarly, we know very little about the degree of gene conversion within *Caenorhabditis* species, which can lead to an increase in homozygosity in the regions of higher recombination [126]. An indirect signature of this process is higher GC content in the regions of higher recombination, which we do indeed observe in these *Caenorhabditis* species [75,131,132]. However, since we currently have no basis for parameterizing these potential effects, the potential for gene conversion was not included in our simulations.

Patterns of genomic diversity with varying patterns of mutational input are uncomfortably similar to those produced by an interaction of selection and recombination alone. It naturally makes sense that regions with more mutation would also have more standing variation. If higher mutation rates are correlated with higher recombination rates, then at least at a superficial level, it would seem to be very difficult to distinguish among these evolutionary forces (Figs 6, S10, S11 and S12). Our evolutionary simulations showed that variation in mutation rate along the genome affects most diversity statistics that we used in this study, except Tajima's D, ω, ZnS, $F_{is}$, and TMRCA. In reality, a complex combination of evolutionary forces influences the genetic variability in these nematodes, which suggests that a deeper understanding of the balance of these forces requires more subtle ways of distinguishing among them, as we attempt to do with our classification analysis below.

## History and structure of populations

Organisms live in dynamic environments that change in space and time, which naturally has the potential to dramatically affect population densities and therefore the context for evolutionary change. *C. elegans* populations locally undergo phases of exponential growth in localized areas of vegetative decay, followed by dispersal to habitats with new resources [52,124,171]. Consistent with this, its global population structure suggests metapopulation dynamics of frequent local extinctions followed by recolonization [93,112,172].

The structure of the local collection of *C. elegans* used here, which is part of a much larger dataset from Hawaii [81], shows both multiple divergent lineages and resampling of a few closely related individuals, consistent with the emerging metapopulation paradigm. And indeed, the inferred demographic history of this population suggests massive reduction and fluctuations in population size over time (Fig 5). While the ancient pattern of population history and size is comparable to the previously reported dynamics in *C. elegans* described by Thomas et al. [71], these demographic histories cannot be directly compared on the recent time scale since we employed genomic data from nematodes isolated from one location, whereas they used pseudo-diploids from a "global" sample.

The global population structure of *C. remanei* similarly implies extensive migration across its range [80,173], but local dynamics are still poorly understood. In the Toronto sample of *C. remanei*, we observed elevated inbreeding coefficient and background linkage disequilibrium along the genome, which is likely the result of within-family mating or similar demographic processes in recent generations. Moving back in time, the inferred evolutionary population

history of this population indicates major fluctuations in the population size of up to two orders of magnitude (Figs 5 and S8A). Even so, the inferred sizes are larger than those predicted for *C. elegans*. One consequence of these estimates is that the total size and overall scale of population dynamics makes it particularly difficult to implement this specific demography in our simulations due to excessive computational demands. Nevertheless, our simulations show that continuous exponential population growth had much stronger effects on most diversity statistics than fluctuations in the population size. Similarly, a recent bottleneck in outcrossing populations, which seems to be the case for the *C. remanei* population, can have a significant impact on most of the diversity statistics used in our study ("outcrossing inbreeding" columns in Figs 6, S10, S11 and S12). Overall, however, these demographic effects are minor relative to persistent differences in the mating system.

A potential explanation for the pattern of diversity within *C. elegans* is that the pattern of polymorphism across the genome is a residual echo of ancestral outcrossing, even in the expected homogenizing effects of self-reproduction. In particular, *C. elegans* is predicted to have switched to selfing reproduction within the last ~4 my [41]. To test this hypothesis, we simulated populations exhibiting a discrete change in the mode of reproduction and then tracked how the diversity decayed over time. We found that polymorphism from the outcrossing ancestor was predicted to decay very rapidly. With that, it is very unlikely that the pattern of diversity that we observe in *C. elegans* is the ghost of past outcrossing and instead that it has been maintained by ongoing low levels of outcrossing within extant populations or as a result of a complex interaction of selection and metapopulation dynamics.

### Exploring evolutionary scenarios through simulations

No complete theoretical framework exists to describe how all of the above factors—selection, drift, mutation, genome organization, mating system, and population history—interact to shape diversity at a genomic scale, despite much theory [36,37,120,122,174,175]. Forward-time evolutionary simulation provides an alternative approach to test hypotheses for complex scenarios designed specifically for the species of interest [8,9,176]. Our simulated diversity landscapes, when compared with the empirical patterns in *C. elegans* and *C. remanei*, confirmed the general conclusion that differences in effective population size were responsible for generating the scale and magnitude of genetic variation, whereas effective recombination and selection coupled with mutation and genome organization, shape the distribution of that diversity across the genome (the genomic landscape of diversity). Thus, changing modes of mating among *Caenorhabditis* species has had a profound effect on both the scale and shape of the diversity landscape.

Evolutionary simulations could potentially be further utilized to evaluate and interpret patterns observed in empirical data by dissecting signatures of different forces and scenarios on genomic diversity, for instance, by using deep-learning methods (see S1 Data Appendix A1). However, it is important to acknowledge that this approach can be computationally challenging and requires control of the quality of input data (genomic coverage and bioinformatic choices) as well as careful consideration of many potential factors such as population history and structure, in addition to evolutionary forces, genome organization, and species biology.

### Conclusion

*Caenorhabditis* nematodes provide a useful model to study evolutionary consequences of selfing, as they have highly divergent mating systems while maintaining an overall similarity in genome organization. Here, we have demonstrated that the recombination landscape of outcrossing *C. remanei* is similar to *C. elegans*, with extensive domains of lower recombination on

the central parts of chromosomes that predict the genomic landscape of the diversity in both species. The scale of genetic polymorphism within selfing *C. elegans* and outcrossing *C. remanei*, however, are dramatically different because of large differences in effective population size and demographic history. These findings support the emerging perspective that understanding patterns of variation at any particular site in the genome requires a global perspective of the forces that shape variation across the genome as a whole.

## Materials and methods

### Genetic map for *Caenorhabditis remanei*

We constructed a genetic map for *C. remanei* from crosses of 2 inbred strains, PX506 and PX553. Initially, *C. remanei* isolates were derived from individuals living with terrestrial isopods (*Oniscidea*) taken from Koffler Scientific Reserve at Jokers Hill, King City, Toronto, Ontario, in October 2008 as described in [177]. Strain PX390 was created from one female mated with 3 males from an isopod. Strain PX393 is from one female and one male from an independent isopod. The strains were propagated for 2–3 generations before freezing. PX506 and PX553 are inbred strains generated from PX390 and PX393, respectively (the parental strain for PX506 was inadvertently specified as PX393 in [75]). To reduce residual heterozygosity, the lines were sib-mated for 28–30 generations before freezing. Nematodes were kept under the standard laboratory conditions according to Brenner [43].

The genetic map was constructed from 4 crosses of *C. remanei* strains PX506 and PX553, (2 crosses of ♀PX506 x ♂PX553 and 2 ♀PX553 x ♂PX506) using parental genotypes and 341 individually sequenced F2 nematodes, all females. Single L4 animals were digested in proteinase K, and the DNA content was linearly amplified with the phi-29 enzyme (GenomiPhiV3, GE life sciences), then normalized samples were processed for bestRAD sequencing [178] with EcoRI restriction site based adapters. Each multiplexed sample set was sequenced on the Illumina Hi-Seq 4000 platform with four lanes of 100 bp paired-end reads (University of Oregon Sequencing Facility, Eugene, OR). Additionally, we sequenced the PX553 parental strain using the Nextera kit (Illumina) and Hi-Seq 4000 platform. The genome sequence of the PX506 strain was generated previously [75].

For parental strains, we checked the sequence quality of reads with FastQC v.0.11.5 [179] and MultiQC v.1.3 [180], trimmed and filtered reads with Skewer v.0.2.2 [181]. The filtered reads were mapped to the *C. remanei* genome (GCA_010183535.1 from the NCBI database) with BWA-MEM v.0.7.17 [182]. Then we filtered reads with SAMtools v.1.5 [183], removed duplicates by Picard tools v.2.17.6 [184], realigned indels, and called variants with GATK v.3.7 and v.4.1 [185]. Variants were filtered with standard GATK hard filters (see discussions in [186–188]), with only diallelic loci being used in the analysis. We also masked repetitive regions, as well as sites with too low or too high of coverage.

To process the bestRAD reads, we filtered reads without barcodes and flipped forward and reversed reads when a barcode was found on the reverse read using Flip2BeRAD [189], demultiplexed reads with process_radtags from the Stacks package v.1.46 [190], followed by an additional adapter and quality trimming with Skewer. Then, we mapped the reads to the *C. remanei* reference genome (GCA_010183535.1) using bwa mem, marked duplicates and recalibrated alignment with variants from the parental, PX506 and PX553, strains with Picard and GATK BaseRecalibrator, filtered reads that did not cover the parental variants, had secondary alignments, or low mapping quality by SAMtools, and called variants with samtools mpileup. We generated the genetic map with Lep-Map3 [191], the order of markers was defined by their positions on the reference genome. The recombination rate per Mb was estimated in R, the boundaries of low and high recombination domains were determined with the pricewise

regression by the segmented package [192] in R. For more details on this part of the analysis, see the scripts at https://github.com/phillips-lab/CR_CE_popgen/tree/main/genetic_map/.

## *C. remanei* and *C. elegans* population genomic data

To study the genomic pattern of diversity in outcrossing nematodes, we sequenced 14 wild individuals of *C. remanei*. Isopods, a phoretic host carrier of *C. remanei* [193], were collected at the same station as the strains used for the genetic map, the Koffler Scientific Reserve in Ontario, Canada, in September 2013, sacrificed within a few hours following collection after having been placed on agar plates seeded with *Escherichia coli* OP50. From each of 14 single *C. remanei* individuals isolated the next day from these samples (S3 Table), genomic DNA was directly amplified using the Repli-G kit (Qiagen), and then sequenced with Illumina HiSeq from TruSeq gDNA libraries by GenomeQuebec. One pair of *C. remanei* individuals derived from a shared isopod host (NS50-1, NS50-2), whereas all other individuals were isolated from different isopods.

To compare the population diversity patterns of this outcrossing species with a selfing species using similar approaches, we re-analyzed genomic sequences of 28 wild isolates of *C. elegans* collected at one location on the Big Island, Hawaii, from Crombie et al. [81]. For more details on the *C. elegans* sample, see Source data 1 from [81]. Sample IDs and the NCBI Sequence Read Archive accession numbers for *C. elegans* and *C. remanei* are in S3 Table.

## Variant calling, diversity statistics, and demography

We filtered and mapped reads to the *C. elegans* genome (project PRJNA13758, from WormBase version WS245) or the *C. remanei* genome (GCA_010183535.1 from the NCBI database) as described for the genetic map parental strains above. The individuals included in this study had an average depth of coverage ranging from 20x to 40x. Variants were filtered with the standard GATK hard filter, only diallelic loci were used in the analysis. Additionally, we masked some genomic regions, such as indels with 10bp flanking regions with GATK and BEDTools v.2.25 [194]; repetitive regions using the masked versions of genomes and a script to extract them from [195]; and regions with too low or high mappability (<5x or >100x coverage in half of the individuals), all masks were combined by BEDTools merge.

We estimated 12 population diversity statistics using diploS/HIC fvecVcf [98] with two minor modifications: allowing not to normalize statistics, and to have only one sub-window in a window (see diploSHIC_note.txt on the project GitHub), and, additionally, the β-statistic using BetaScan [196] for 100 kb windows in *C. elegans* and *C. remanei* samples. To estimate the β-statistic for empirical data, we first applied an imputation with beagle v.5.0 [197] and a data format conversion by glactools [198]. Nucleotide diversity within introns and exons was estimated using these features from corresponding genome annotations, VCFtools [199], and BEDTools. We compared nucleotide diversity between domains of recombination and different gene features using Cohen's d from the lsr package [200] and the Fisher-Pitman Permutation Test (Z) from the coin package [201] in R [202].

Demographic history was inferred by SMC++ v1.15.1 [113] for 100 bootstrapped replicates of 8 individuals using data from chromosomes I, III, IV for *C. elegans* and I, II, III, V for *C. remanei*. Additionally, we performed this analysis for all chromosomes separately. The genomic regions with unfit mappability or repeats were masked as described above. We used the mutation rate of 2.3e-09 base substitution per generation [106], and, for *C. elegans*, we rescaled this mutation rate by 0.5 and, later, the obtained generation time by 2 due to selfing (as discussed in [203]). To compare estimates of the recent population sizes from SMC++, as well as with previous estimates within *C. elegans*, we also calculated effective population size from

nucleotide diversity data using the Watterson estimator [204] assuming neutrality and complete selfing for *C. elegans* (see [26,205], Ne = N/2).

The inbreeding coefficient ($F_{is}$) along the genome was estimated for sites with a minor allele frequency of more than 0.05 using VCFtools v.0.1.15, BEDtools v.2.25, HTSlib v.1.6, plink v1.90b4.6 [206], and popStats v1.0.0 [207]. The sample structure was visualized in the two-dimension space using latent coordinates from popVAE [208]. The effective recombination rate along the genomes was inferred by ReLERNN [176], using one of the replicates of reconstructed demography from SMC++ and (x0.5) mutation and (x2) time scaling for the selfing species. Linkage disequilibrium within and across chromosomes and the LD decay were estimated with plink using all sites for *C. elegans* sample, and every 10th site for *C. remanei* for all LD calculations except the fine-resolution LD where we used all sites.

Figures were plotted via R packages dichromat [209], ggplot2 [210], gridExtra [211], ggpubr [212], magick [213], and also boot [214], coin [215], lsr [216], reshape2 [217], scales [218] to estimate summary statistics. The scripts for variant calling and masking, diversity statistics estimation, demography reconstructions, populations structure, effective recombination inference, LD decay, and corresponding figures and statistics are in https://github.com/phillips-lab/CR_CE_popgen/tree/main/diversity_stats/.

### Reconstruction of the ancestral states of *C. remanei*

We reconstructed the ancestral states of the *C. remanei* and *C. latens* reference genomes. We used genomes of four strains of C. remanei, PX506, PX356 (GCA_001643735.2), PB4641 (GCA_000149515.1), PX439 (GCA_002259225.1) and *C. latens* (GCA_002259235.1). Preliminary topology was obtained with progressiveMauve [219] with 1 Mb regions from each of 6 chromosomes. Next, the genomes were masked based on their mappability by GenMap [220], and aligned by Progressive Cactus [221] with the following species tree topology: (((((*C. remanei* PB4641, *C. remanei* PX356), *C. remanei* PX439), *C. remanei* PX506), *C. latens*). The ancestral states were re-estimated by ancestorsML tool in the HAL tools [222].

Then for each chromosome, we calculated the fractions of sites with ancestral states, GC content in the ancestor, various types of substitutions per 100 kb non-overlapping windows, and extracted ancestral states positions polymorphic in the *C. remanei* population by HAL tools, BEDtools, and bash. We used Relate [114] to infer the history of the *C. remanei* population based on the ancestral states and the recombination map described above. We also examined demographic tests of the signal of positive selection along the genome, adjusting reported p-values using the harmonic mean p-value within 3 Mb sliding windows (2 Mb overlap) from harmonicmeanp [223] R package. The time to the most recent common ancestor (TMRCA), the relative TMRCA half-life (RTH, following [224]), and terminal branch lengths were assessed by phangorn [225] and phytools [226] R packages from Relate generated trees. Statistical analysis and visualization was made with coin, data.table [227], dplyr [228], ggplot2, gridExtra, ggpubr, lsr, pals [229], and scales packages in R. All scripts for the ancestral reconstruction, demographic inference, and related analyses available at https://github.com/phillips-lab/CR_CE_popgen/tree/main/ancestral.

### Evolutionary simulations

To understand how various factors affect the genomic landscape of diversity, we run forward-time individual-based evolutionary simulations in SLiM v.3.3 [8,9] using the tree-sequence format [10]. We performed three types of simulation: 1) effects of the selfing rate, selection, and mutation landscape; 2) the decay of the ancestral diversity; 3) effects of rapid changes in population size on estimated statistics. All simulations had a population size of 5,000

individuals, 1 chromosome of 3 Mb in size with three recombination domains: left and right arms (1 Mb) with a high recombination rate (2.5e-7), and a 1 Mb central domain of low recombination (1e-9) to mimic the recombination domains of lower and higher recombination in nematodes.

In the first type of simulations, we changed selfing, mutation landscape, and selection regime. The selfing rate was 0% (outcrossers), 90%, 98%, 99.9%, or 100% (selfers). The mutation landscape was uniform with the mutation rate of 2e-8 in all domains of recombination. While this is an order of magnitude higher than actual mutation rate estimates, higher rates greatly facilitate the simulation process and, since the emphasis is on the relative values of mutation on arms and centers, this difference should not affect the normalized statistics used above. We used several domain-specific patterns of mutation rate differences: a uniform landscape with no difference in the mutation rate on the arms and the central domain (denoted as 1-1-1), 15% more mutations on the arms than on the centers (1.15-1-1.15), 50% more (1.5-1-1.5), or 200% more mutations on the arms (2-1-2), the 1x mutation rate was the same in all simulations. Slightly elevated mutation rates within regions of higher recombination have been shown for nematodes previously [32,49]. We considered four main types of selection regimes: only neutral mutations; neutral and 10% deleterious mutations; neutral, 10% deleterious, and 1% beneficial mutations; neutral, 10% deleterious, and 1% balancing mutations. Non-neutral simulations utilized various distributions for selection, and simulations with balancing selection used only one of these distributions (see details in S4 Table). Distributions of dominance were different for deleterious and beneficial mutations, with a shift towards recessive for the former and more additive coefficients for the latter (see S14 Fig and the SLiM scripts). All simulations were run for 50,000 generations (10N, burn-in).

The second type of simulations was designed to explore the effects of a switch from ancestral outcrossing to selfing. We ran neutral and non-neutral simulations (see details in S4 Table) in which all populations were outcrossing during burn-in and then subsequently changed to either 98 or 100% selfing. To observe the decay of ancestral diversity we saved results every 5,000 generations for 30,000 (6N) generations and repeated each scenario 50 times.

The aim of the third type of simulations was to reveal the consequences of rapid change in population size on the diversity statistics used in our analyses. These simulations utilized only neutral mutations but allowed different selfing rates and mutational landscapes. First, we performed simulations with 100 generations of exponential growth of 3% following burn-in, generating a rapid population size increase from 5K to about 100K. Second, we investigated the effects of fluctuations in population size following burn-in, by setting the population size to 15,000 and then back to 5,000 for 10 generations and so on 5 iterations. For the third type of scenario with exponential growth or fluctuations of the population size, we compared diversity statistics for the same populations at the burn-in generation and at the end of the simulation for 50 replicates.

We added neutral mutations, "recapitated" trees, and converted tree sequences from SLiM simulations to the VCF file format with 100 individuals using tskit v.0.3.4 [230], msprime v.0.6.1 [231], pyslim [232], and estimated diversity statistics with diploS/HIC and BetaScan as for empirical data but for 40 kb windows with and without normalization. We also estimated the divergence from the ancestral genome using bash and BEDtools, $F_{is}$ statistics via popStats, and calculated tree heights (TMRCA) from tree-sequences using python modules argparse [233], msprime, statistics [234], and pyslim.

We plotted and analyzed simulation results using packages coin, corrplot, dichromat, dplyr, ggplot2, gridExtra, ggpubr, and lsr packages in R. SLiM scripts for simulations, and related R scripts for visualization and analysis are in https://github.com/phillips-lab/CR_CE_popgen/tree/main/simulations/. Additionally, we performed a classification of simulated and empirical data by evolutionary scenarios. The details of this approach are described in S1 Data Appendix A1.

## Supporting information

**S1 Fig. The number of crossover events in F2 individuals from 4 crosses of *C. remanei* inbred lines.** A1 and A2 are crosses of ♀ PX506 x ♂ PX553; whereas B1 and B2 are crosses of ♀ PX553 x ♂ PX506. We observed similar distributions of the number of crossover events in all crosses. Autosomes had more recombination events than the sex chromosome (X).
(TIF)

**S2 Fig. Genomic diversity statistics in *C. elegans* and *C. remanei* populations.** Dots represent the diversity statistics estimated in 100 kb non-overlapping windows, whereas lines show locally weighted smoothing of these values. Windows with less than 10% covered positions were removed from the analysis. The vertical dashed lines indicate the boundaries of regions of low recombination central domain. The x-axis represents the normalized genome position. See the description of statistics in the Methods section. In almost all statistics, *C. remanei* and *C. elegans* exhibit distinct patterns and scales.
(TIF)

**S3 Fig. Landscape of nucleotide substitutions in the *C. remanei* PX506 reference genome, as estimated from inferred ancestral states.** The first two lines (dark teal and teal) are transitions and the other lines are various forms of transversions. **(A)** Percent of substitutions for each class as estimated from ancestral GC content as a fraction of coverage of a 1 Mb genomic window. **(B)** Relative fraction of each substitution type at a given genomic location. Overall, relative proportions for three of the substitution types are homogeneous along the genome, while the three other types (C→T|G→A, A→T|T→A, and C→G|G→C, marked with asterisks) show small but significant differences between domains of recombination (see S2 Table).
(TIF)

**S4 Fig. Two-dimensional representation of relatedness and structure in *C. remanei* and *C. elegans* populations.** LD1 and LD2 show two latent dimensions. Some individuals in *C. remanei* population are closely related. In the *C. elegans* population, there are few lines with several individuals that are almost genetically identical and were combined to isotypes in previous studies ([81], see S3 Table).
(TIF)

**S5 Fig. Genome wide patterns of linkage disequilibrium.** The panels show linkage disequilibrium ($r^2$) in the *C. elegans* **(A)** and *C. remanei* **(B)** populations. The linkage between and within chromosomes is highly similar in *C. elegans*, but significantly different (see the main text). *C. remanei* shows the fast decay of linkage disequilibrium (Fig 4) and low interchromosomal LD.
(TIF)

**S6 Fig. Genome-wide landscape of recombination inferred from population diversity data of *C. elegans* and *C. remanei*.** The x-axis shows the normalized genome position. The vertical dashed lines indicate the boundaries of central regions of low recombination obtained from genetic maps.
(TIF)

**S7 Fig. Demographic history of populations of *C. elegans* and *C. remanei* inferred from each chromosome.** The color represents chromosomes. We ran 100 bootstrapped replicates using eight individuals from each species, each line represents one replicate. The grey shadow indicates the region of recent demographic history, where estimations are less accurate. We used one generation per year in this analysis and scaled of the mutation rate (x0.5) and

coalescent time (x2) for *C. elegans*.
(TIF)

**S8 Fig. Analysis of inferred genome-wide genealogies of the *C. remanei* population. (A)**
Demographic history of the *C. remanei* population estimated for each chromosome. **(B)** Tree
statistics calculated from the genealogies averaged for 100 kb windows; TMRCA (time to the
most recent common ancestor), RTH (relative TMRCA half-time), and the lengths of terminal
branches of the trees. **(C)** Signatures of positive selection along the genome, the y-axis shows
the p-values after the correction on multiple comparisons using the harmonic mean approach
(see the Methods). **(D)** Quantile-quantile plot displays p-values from the tests for positive
selection (y-axis) versus the expected uniform distribution of p-values (x-axis). The yellow
color shows sites on the arms, and the black color indicates sites on the central parts of chromosomes.
(TIF)

**S9 Fig. The distribution of selection and dominance coefficients of beneficial and deleterious mutations in outcrossing simulated populations.** This picture depicts mutations from
the SD&SB-SD&SB class described in S4 Table with the uniform mutation landscape. **(A)** Percentage of mutation classes with allele frequency more than 0.5 at the beginning of the simulation ("Initial") and the end on the arms and centers. Colors display the class of dominance
coefficient (h), and the columns represent the strengths of selection (absolute values of Ns,
where N is the population size of 5,000 and s is the selection coefficient). **(B)** The percent of
corresponding mutation classes of mutations with allele frequency less than 0.5.
(TIF)

**S10 Fig. Distributions of diversity statistics in simulated populations.** Lines represent
locally weighted smoothing of the values estimated per 40 kb non-overlapping windows, the
vertical dashed lines indicate the boundaries of central domain with low recombination rate.
Columns show the outcrossing rate, where "outcrossing" means completely outcrossing populations and, in other columns, % specify the percentage of selfing in population; "outcrossing
inbreeding" corresponds to scenarios with outcrossing populations that underwent the bottleneck at the very end of simulations (see Methods). Rows represent domain-specific differences
in mutation rate, with 1-1-1 is the uniform mutation landscape, 1.15-1-1.15 means 15% more
mutations on the arms, 1.5-1-1.5 is 50% more mutations in domains of high recombination,
and 2-1-2 means two times more mutations in domains of high recombination. Colors show
the selection regime (see details in Methods). On this figure, shown only 4 selection regimes
that are specified with asterixis in S4 Table.
(TIF)

**S11 Fig. Distributions of diversity statistics in simulated populations for scenarios with
neutral and deleterious and beneficial mutations.** See description to S10 Fig and parameters
in S4 Table.
(TIF)

**S12 Fig. Distributions of diversity statistics in simulated populations for scenarios with
neutral and deleterious mutations.** See description to S10 Fig and parameters in S4 Table.
(TIF)

**S13 Fig. Differences in diversity statistics between simulations with shifts in the population size compared to the values of the statistics of the corresponding simulation before
the changes in neutral scenarios.** The colors represent the fold change in statistics at the end
of the simulation versus before changes in size. **(A)** Fluctuation in population size for 100

generations, where every five generations, the population size went from 5,000 to 15,000 and then back. **(B)** Exponential growth of 3% for 100 generations.
(TIF)

**S14 Fig. Distributions of selection and dominance coefficients used in evolutionary simulations.** The columns show the percentage of each class of selection coefficients drawn from gamma distributions with different parameters (see S4 Table). Dominance coefficients were chosen independently from a mixture of uniform and beta distributions with distinct parameters for deleterious mutations and beneficial mutations (see the Methods and SLiM scripts at https://github.com/phillips-lab/CR_CE_popgen/tree/main/simulations).
(TIF)

**S1 Table. Supplementary information for the *C. remanei* genetic map.**
(XLSX)

**S2 Table. The difference in the relative fraction of substitutions from the *C. latens* and *C. remanei* common ancestor to the *C. remanei* strain PX506 between "arms" and "centers".**
(XLSX)

**S3 Table. Individually sequences worms used in this study.**
(XLSX)

**S4 Table. Distribution of selection coefficients used in evolutionary simulations.**
(XLSX)

**S1 Data Appendix A1. Classification of simulated and empirical data using convolutional neural networks.** We describe an approach to classify evolutionary simulations and empirical data by evolutionary scenarios using deep learning, its application, caveats and future directions.
(PDF)

## Acknowledgments

We thank Rose Reynolds, Timothy Ahearne, Chadwick Smith, Scott Scholz for *C. remanei* mapping strain generation, Larry Meng for DNA isolation of *C. remanei* nematodes from the Koffler Scientific Reserve used for population genomic sequencing. Many thanks go to Sean O'rourke and members of Mike Miller's lab at UC Davis for help with bestRAD sequencing, as well as to Gavin Woodruff, Murillo Rodrigues, Peter Ralph, Andy Kern, and other former and current members of Phillips lab and Kern-Ralph co-lab at the University of Oregon for comments and discussions. We thank the Research Advanced Computing Services team for assistance with the computing cluster Talapas at the University of Oregon and the Genomic Core Facility at the University of Oregon for assistance with library preparation.

## Author Contributions

**Conceptualization:** Anastasia A. Teterina, Richard Jovelin, Asher D. Cutter, Patrick C. Phillips.

**Data curation:** Anastasia A. Teterina, John H. Willis, Asher D. Cutter.

**Formal analysis:** Anastasia A. Teterina, Matt Lukac.

**Funding acquisition:** Asher D. Cutter, Patrick C. Phillips.

**Investigation:** Anastasia A. Teterina, John H. Willis, Matt Lukac, Asher D. Cutter.

**Methodology:** Anastasia A. Teterina, John H. Willis, Patrick C. Phillips.

**Project administration:** Asher D. Cutter, Patrick C. Phillips.

**Resources:** Richard Jovelin, Asher D. Cutter, Patrick C. Phillips.

**Software:** Anastasia A. Teterina, Matt Lukac.

**Supervision:** Asher D. Cutter, Patrick C. Phillips.

**Validation:** Anastasia A. Teterina, Matt Lukac, Patrick C. Phillips.

**Visualization:** Anastasia A. Teterina, Matt Lukac.

**Writing – original draft:** Anastasia A. Teterina, Patrick C. Phillips.

**Writing – review & editing:** Anastasia A. Teterina, John H. Willis, Matt Lukac, Richard Jovelin, Asher D. Cutter, Patrick C. Phillips.

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
