## [Decision Letter · Decision Letter 0]

6 Mar 2023

Dear Dr Teterina,

Thank you very much for submitting your Research Article entitled 'Genomic diversity landscapes in outcrossing and selfing *Caenorhabditis* nematodes' to PLOS Genetics.

The manuscript was fully evaluated at the editorial level and by independent peer reviewers. The reviewers appreciated the attention to an important problem, but raised some substantial concerns about the current manuscript. Based on the reviews, we will not be able to accept this version of the manuscript, but we would be willing to review a much-revised version. We cannot, of course, promise publication at that time.

If you decide to revise the manuscript for further consideration at PLOS Genetics, please aim to resubmit within the next 60 days, unless it will take extra time to address the concerns of the reviewers, in which case we would appreciate an expected resubmission date by email to plosgenetics@plos.org.

We are sorry that we cannot be more positive about your manuscript at this stage. Please do not hesitate to contact us if you have any concerns or questions.

Yours sincerely,

Ian R. Henderson

Academic Editor

PLOS Genetics

Kirsten Bomblies

Section Editor

PLOS Genetics

Reviewer's Responses to Questions

**Comments to the Authors:**

Reviewer #1: Review of Genomic diversity landscapes in outcrossing and selfing *Caenorhabditis* nematodes

Anastasia A. Teterina1,2, John H. Willis1, Matt Lukac1, Richard Jovelin3, Asher D. Cutter3, Patrick C. Phillips1

General

I really liked this MS and think the data and the analyses are (in large measure) important and well performed. I have an issue with the AI modelling component which seems under-described and under-performs.

The core findings - that genome-wide diversity, LD and related measures are all indicative of higher effective population sizes in CaeRema, and that the arms-and-centres structure identified in CaeEleg is present in CaeRema and is likely driven by the same recombinational landscape - are important, and show the impact of selfing on CaaEleg and especially the impact of linked selection/sweeps. These findings are not _unexpected_ as single genome assemblies of several Caenorhabditis [and other] species have already shown similar strong patterning on chromosomes, but they are significantly deepened by the population genomic analysis of CaeRema. While this is currently a 2-way comparison (and thus could be significantly biased by unknown factors impacting either species) the analyses are well performed and - given that they conform to much theoretical inference - are likely to be very strong.

One of the critical findings from the work of the Andersen lab recently has been the discovery and definition of a large number and span of highly divergent regions in CaeEleg where short reads do not map. This generates a false run of homozygosity against the reference if the reference region(s) are present, or an uncovered gap if alternative regions are present This structure is significant (up to 20% of the genome is affected.) Is similar structure present in the CaeRema cross parents, or in the resampling data (perhaps particularly in the divergent individuals)? If present, defining it might assist in partitioning analysis of variation to only deal with regions of the reference that can be/are expected to be covered, and thus can have credible allele calls. The comment at line 969 makes me wonder if just this process might be at play here.

The modelling is interesting in showing the interacting impacts of different population level processes (neutral and adaptive) in patterning the diversity along chromosomes, and is a useful extension of the inferences drawn from the population genomic data. The statement "as predicted by theory" rings loud here - what new or surprising is present in these analyses and what does the contrast between CaeRema and CaeEleg bring to the discussion. The text here could be shortened significantly in this context.

The convolutional neural network component of the work is the weakest and I would suggest that it is removed from the MS. As I understand it the authors used a set of parameters derived from the empirical data to generate a set of synthetic data on a set of model chromosomes, and then back tested the emergent classifier on the real data. The classifier performed poorly, suggesting that the real CaeRema data derived from a selfing population rather than a sexual one. While it is important to publish negative results, I am not convinced that these constitute real negative data - they suggest that the CNN was not able to extract enough information from the model chromosomes which in turn suggests the modelling was not modelling real world processes (as we assume that the measurement of CaeRema biology - ie outcrossing predominates in a some what closed and thus inbred population - is correct).

Analytic comments

The CaeRema and CaeEleg genomes are different sizes (~25%) and it is thus of interest to ask where this difference lies. This analysis impacts on analyses that compare the "proportion" of each chromosome that is in each partition (arm; centre; arm). My calculations indicate that the centre partitions in both species have very similar spans, and that the expansion in CaeRema [or shrinkage in CaeEleg) is largely due to change in the arm partition spans. In the context of discussions of what drives the differences in patterns of evolution between the species this is I think germane.

Similarly, it would be good to know the arms vs centres distribution of the ~56% of the CarRema genome for which ancestral states could be inferred: my suspicion is that these 56% are biased to the centres, and this might impact analyses (though from Fig 3 the contrasts are so clear that I suspect that any impact might be minimal).

In lines 278ff it is mentioned that a "clearer definition" of the boundaries of arms and centres was possible. Does this mean that the numerical base position of the boundaries was changed because of this "clarification"? If not, it is not clear what clarification was afforded.

lines 308ff It is not clear what analysis is being reported. Is this the reference genome vs the imputed LCA? Do the authors have an explanation for the strong difference in tr/tv ratios? Why would the between species differences be so different from the within species ones for this metric? I would have liked to see a discussion of the impact of using a population of CaeRema and a single reference CaeLate in imputing the ancestor: is this the source of the difference?

Minor questions

Tab3 line 398

What does "low accuracy" mean in the methods column? Is it a statement on the performance of the method class as a whole (in which case remove) or a particular form of coalescent analysis (in which case we need a definition).

Textual comments

There are several textual infelicities in the MS which make reading somewhat difficult in places. Some are (imho) grammatically wrong, others I think obscure the meaning and content.

I found the opening paragraphs of the introduction over-general, and sometimes hard to parse scientifically. This could be shortened and focussed. The general statement that there are both selective and neutral processes at work, and that these processes overlap with each other in their impacts on the genome, and that general structural features of the genome - driven by selection on structure in general - can have consequences for/impose constraints on realise change, is unproblematic... but is more textbook introduction than research paper introduction.

eg line 48 "factors act and constantly interact"? Not sure of the meaning of this phrase.

The authors close the legends of (most) figures with a narrative discussion of what the figure is being used to show. My style is to leave the figures to show data and to have such narrative in the main text. I am not sure what PLoS protocol is.

Other text / presentation notes

* arms and centres are defined twice, and the definitions suggests capital-A Arms and capital C-Centres are the short forms (which are not used again).

* I would suggest moving the lists of values/means, p-values, SDs etc from section "Genetic diversity.." [lines 186ff: lines 194ff, 208ff, 222ff, 227ff] to a table where the hypothesis tested, the statistical test and the list of values could be presented accessibly. The current extended sentence listing all values is hard to read and obscures the statement of the actual findings. I would also suggest using the same exponent for all measures within a set/comparison (eg if values are 1.2 x10-2 and sd is given as 2.3 x 10-3 thuis suggests greater accuracy/significant figures for the SD. Give sam number sig figures for each, and think about normalising between species where one is x10-2 and the other x10-3)

* the plots of metrics across the genomes are stated to be measured in sliding windows of 100 kb (mostly) [eg Fig 2, Fig 3]. Is this really a sliding window? in which case what were the step sizes? or is it a tiled window of 100 kb (which is equivalent to a sliding window of 100k with a step of 100k, but, I would contend is categorically different - no sites are counted > once in a step window, whereas they are in a sliding window)

* I have a campaign to halt the use of the word "worm" when nematode is correct (and there are so many other worm-like taxa). Perhaps the authors could look at the ~9 instance of the use of worm(s) and decide if nematode(s) would be better.

* spans are sometimes presented as "100-kb", and sometimes as "100 kb". I think 100 kb is correct in each case. There are a few instances of no separation between value and unit

* sometimes "ten times" is spelt out, sometimes it is given "10x" [etc]; use the spelt out version

* line 232 "recombination rate IN"

* line 239 location of "details" missing

* occasional flipping between past tense and present tense in reporting of work done / results found (even in same sentence) - see lines 354ff for eg.

* line 340 is this a new paragraph, or just a misplaced carriage return?

* line 440 data "show" (data are plural)

Reviewer #2: Taking advantage of the Caenorhabditis clade, where selfing has evolved independently three times, Teterina et al use genomic analyses and evolution simulations to provide insight into the contribution that selfing vs. outcrossing has on genetic diversity. The authors find that while the overall pattern of diversity mirrors the underlying pattern of genetic recombination on chromosome arms vs. centers in both C. elegans (selfing) and C. remanei (outcrossing), C. remanei has significantly higher levels of genetic diversity as is predicted for an outcrossing species. This is an important study that sets the stage for future work to elucidate contributions of reproduction, selection and demography to genetic diversity. The results will be of interest to the broad audience of PlosGenetics.

1. The authors should consider making the manuscript more assessable to those outside population biology – they often mention programs, parameters, and values that have little meaning to those outside the field.

2. The authors should provide more information in the introduction about what is known about selfing vs. outcrossed species. There are some very interesting genetic studies from the Haag, Schedl and Ellison labs that show only a couple of mutations are required to convert between these two different reproductive modes.

3. I understand that this is outside the scope of this study, but it seems a more powerful approach would be look at more than one self vs. obligate outcrossing species to see whether the differences observed here are more generalizable. In the absence of this, the authors should be a little more careful in not generalizing their results beyond the two strains they examined.

Reviewer #3: This paper provides some important findings for C. remanei, including a genetic map and a polymorphism/LD survey of natural isolates. The species is of broad interest because it is an outcrossing relative of C. elegans, and because of the strikingly high level of nucleotide diversity found in natural populations. The authors also pair the new data with a reanalysis of existing data for a C. elegans population, and the contrast between the two is informative. Overall the genetic maps between the two species are qualitatively similar, and the drastic polymorphism and LD differences between the species (which were previously discovered), are consistent with the major life style difference (i.e. remanei is an outcrossing species whereas elegans reproduces primarily by self-fertilization).

One significant puzzle is not explained in the paper, and really needs to be addressed more fully. The inbreeding coefficient for C. remanei was found to be 0.38, a strikingly high figure for an outcrossing species. The PC plot (fig S4) does not really help us understand the cause of this, and neither does the corresponding text “ This cluster was most likely formed from individuals from a single-family lineage displaying intensive”.

There are a couple things about the Fis value that need attention here. (1) Are there runs of autozygosity consistent with consanguineous mating? (2) How long ago did the inbreeding occur? (3) How could inbreeding have occurred if the individuals mostly came from different isopods? (4) Were the sampled isopods from different species?

Lastly, regarding the high Fis, could there be a sequencing or pipeline problem that missed a lot of heterozygotes? The methods are thin regarding this possibility, and appear to not even tell us what the read depth was (sorry if I missed it somehow).

The analysis of changing population size on pp 16-18 and figure 5, is confusing. The two parts of figure 5 seem to tell different stories over recent times, and in particular the SMC++ suggests a flat history for everyone back to 1000 generations ago (albeit with a wide variance depending on the replicate). This is not consistent with the LD based story, nor the text. On balance it seems these analyses just are not working well, and it might be best to just report the conflicting signals, along with a figure of the SFS.

Unfortunately, the simulations and the neural network analyses add very little to the paper. The problems here start with the motivating rationale, which was to simulate data under a wide array of population genetic models to look for the kinds of model differences that mirror the observed differences seen between the elegans and remanei data. This kind of approach, without specific questions, and without much use of data driven estimators, seems bound to provide results that are hard to interpret.

**Have all data underlying the figures and results presented in the manuscript been provided?**

Reviewer #1: Yes

Reviewer #2: Yes

Reviewer #3: Yes

PLOS authors have the option to publish the peer review history of their article (what does this mean?). If published, this will include your full peer review and any attached files.

Reviewer #1: **Yes: **Mark Blaxter

Reviewer #2: No

Reviewer #3: No

---

## [Decision Letter · Decision Letter 1]

5 Jun 2023

Dear Dr Teterina,

Thank you very much for submitting your Research Article entitled 'Genomic diversity landscapes in outcrossing and selfing *Caenorhabditis* nematodes' to PLOS Genetics.

The manuscript was fully evaluated at the editorial level and by independent peer reviewers. The reviewers appreciated the attention to an important topic but identified some concerns that we ask you address in a revised manuscript.

We therefore ask you to modify the manuscript according to the review recommendations. Your revisions should address the specific points made by each reviewer.

Yours sincerely,

Ian R. Henderson

Academic Editor

PLOS Genetics

Kirsten Bomblies

Section Editor

PLOS Genetics

Reviewer's Responses to Questions

**Comments to the Authors:**

Reviewer #2: The revised manuscript by Teterina et al presents genetic and genomic analyses and evolution simulations to provide insight into the contribution that selfing vs. outcrossing has on genetic diversity. The authors find that while the overall pattern of diversity mirrors the underlying pattern of genetic recombination on chromosome arms vs. centers in both C. elegans (selfing) and C. remanei (outcrossing), C. remanei has significantly higher levels of genetic diversity as is predicted for an outcrossing species. Evolutionary simulations provide insight into contributions of selection, recombination, mutation, and selfing on genetic variation. The authors have done a good job addressing the previous reviews and the results will be of interest to the broad audience of PlosGenetics.

Reviewer #3: This paper is improved.

However I remain concerned about the estiamtes of historic population size, and apparent inbreeding.

There seems to be a good chance that something fairly fundamental about the history of the sampled population has been overlooked. There are two main clues to this. First are the analyses of historic population size. As mentioned previously, the two kinds of analyeses shown in the figure have little in common. Both can’t be right. Coloring figure 5 does little to alleviate the discrepancy.

Second, there is the high level of Fis. The author’s suggest recent consainguineous mating, which could certainly do it. However this would also leave a high variance in Fis along the chromosomes, something which was not observed. Again, something is amiss.

The paper is informative on the mapping and polymorphism fronts, but the population genetic analyses are incomplete.

**Have all data underlying the figures and results presented in the manuscript been provided?**

Reviewer #2: Yes

Reviewer #3: Yes

PLOS authors have the option to publish the peer review history of their article (what does this mean?). If published, this will include your full peer review and any attached files.

Reviewer #2: No

Reviewer #3: No

---

## [Editor Report · Decision Letter 2]

21 Jul 2023

Dear Dr Teterina,

We are pleased to inform you that your manuscript entitled "Genomic diversity landscapes in outcrossing and selfing *Caenorhabditis* nematodes" has been editorially accepted for publication in PLOS Genetics. Congratulations!

Yours sincerely,

Ian R. Henderson

Academic Editor

PLOS Genetics

Kirsten Bomblies

Section Editor

PLOS Genetics

Comments from the reviewers (if applicable):

**Data Deposition**

http://datadryad.org/submit?journalID=pgenetics&manu=PGENETICS-D-22-01473R2

**Press Queries**

---

## [Editor Report · Acceptance letter]

10 Aug 2023

PGENETICS-D-22-01473R2 

Genomic diversity landscapes in outcrossing and selfing *Caenorhabditis* nematodes

Dear Dr Teterina, 

We are pleased to inform you that your manuscript entitled "Genomic diversity landscapes in outcrossing and selfing *Caenorhabditis* nematodes
 " has been formally accepted for publication in PLOS Genetics! Your manuscript is now with our production department and you will be notified of the publication date in due course.

With kind regards,

Marianna Bach

PLOS Genetics

On behalf of:
